# TGF-β signaling and Creb5 cooperatively regulate Fgf18 to control pharyngeal muscle development

**Jifan Feng, Xia Han, Yuan Yuan, Courtney Kyeong Cho, Eva Janečková, Tingwei Guo, Siddhika Pareek, Md Shaifur Rahman, Banghong Zheng, Jing Bi, Junjun Jing, Mingyi Zhang, Jian Xu, Thach-Vu Ho, Yang Chai\***

Center for Craniofacial Molecular Biology, University of Southern California, Los Angeles, United States

**Abstract** The communication between myogenic cells and their surrounding connective tissues is indispensable for muscle morphogenesis. During late embryonic development in mice, myogenic progenitors migrate to discrete sites to form individual muscles. The detailed mechanism of this process remains unclear. Using mouse levator veli palatini (LVP) development as a model, we systematically investigated how a distinct connective tissue subpopulation, perimysial fibroblasts, communicates with myogenic cells to regulate mouse pharyngeal myogenesis. Using single-cell RNAseq data analysis, we identified that TGF-β signaling is a key regulator for the perimysial fibroblasts. Loss of TGF-β signaling in the neural crest-derived palatal mesenchyme leads to defects in perimysial fibroblasts and muscle malformation in the soft palate in *Osr2*Cre*;Tgfbr1*fl/fl mice. In particular, Creb5, a transcription factor expressed in the perimysial fibroblasts, cooperates with TGF-β signaling to activate expression of *Fgf18*. Moreover, Fgf18 supports pharyngeal muscle development *in vivo* and exogenous Fgf18 can partially rescue myogenic cell numbers in *Osr2*Cre*;Tgfbr1*fl/fl samples, illustrating that TGF-β-regulated Fgf18 signaling is required for LVP development. Collectively, our findings reveal the mechanism by which TGF-β signaling achieves its functional specificity in defining the perimysial-to-myogenic signals for pharyngeal myogenesis.

**\*For correspondence:** ychai@usc.edu

**Competing interest:** The authors declare that no competing interests exist.

## Editor's evaluation

The authors bioinformatically analyze previous scRNA-seq datasets of the developing mouse soft palate to identify differential signaling pathway activities in the heterogeneous palatal mesenchyme. Identifying TGF-β signaling pathway activity with the perimysial cells, they hypothesize and test whether TGF-β signaling in the perimysial cells might regulate palatal muscle formation. This paper will be of high interest to developmental biologists interested in the molecular regulation of tissue interactions that occur during mammalian palate morphogenesis.

## Introduction

Inductive tissue-tissue interaction constitutes a key mechanism for organ development and tissue homeostasis (*Souilhol et al., 2016*; *Takahashi et al., 2013*; *Vainio and Müller, 1997*). The interaction and communication of myogenic cells with the surrounding connective tissues are also indispensable for muscle morphogenesis. Although the detailed cellular and molecular aspects of this dialogue remain to be elucidated, the connective tissues have been shown to serve as a signal source and 'pre-pattern' for muscle differentiation and patterning during development (*Michailovici et al., 2015*; *Nassari et al., 2017*).

At the onset of pharyngeal muscle development, myogenic progenitors migrate from cranial paraxial mesoderm into the center of each pharyngeal arch to form the primary myogenic sites surrounded by cranial neural crest (CNC)-derived cells. Later, in coordination with the development of other CNC-derived craniofacial connective tissues, myogenic progenitor cells in these mesodermal cores migrate to the final myogenic sites while being segregated and patterned into discrete muscle masses (*Noden and Francis-West, 2006*; *Sambasivan et al., 2011*; *Shih et al., 2008*; *Ziermann et al., 2018*). A distinct regulatory network upstream of myogenesis has been identified during early-stage pharyngeal muscle development (*Buckingham and Rigby, 2014*; *Sambasivan et al., 2011*), but the regulatory mechanism that establishes the fine-tuned craniofacial muscle anlagen in their final myogenic sites remains unknown (*Noden and Francis-West, 2006*).

In the final stage of pharyngeal muscle formation, the first and second pharyngeal arch mesodermal cores respectively contribute to the muscles responsible for mastication and facial expression; the muscles in the oropharyngeal region crucial for breathing, swallowing, and speaking are derived predominantly from the fourth pharyngeal arch, including all the pharyngeal constrictors and soft palatal muscles except the tensor veli palatini (*Frisdal and Trainor, 2014*; *Li et al., 2019*; *Michailovici et al., 2015*; *Shiba and Chhetri, 2019*; *Sugii et al., 2017*). Notably, interactions between CNC-derived and myogenic cells become more prominent as the CNC-derived cells start to contribute to connective tissues while individual muscles segregate and proceed to their final locations (*Ziermann et al., 2018*).

During this late pharyngeal muscle development in mice, most pharyngeal muscles of the head have already segregated and reached their ultimate locations between E11 and E13.5 (*Noden and Francis-West, 2006*). In contrast, the soft palatal muscle anlage development starts later, concurrent with the soft palatal shelf primordium formation (*Han et al., 2021*; *Li et al., 2019*). From E13.5 onwards, individual soft palatal muscles, particularly the levator veli palatini (LVP), can be clearly detected segregating from the rest of the pharyngeal muscles and populating the soft palate primordium, patterning and migrating towards the midline in parallel to the lateral-to-medial outgrowth of the connective tissues consisting of the CNC-derived palatal mesenchymal cells (*Grimaldi et al., 2015*; *Han et al., 2021*; *Li et al., 2019*). Therefore, soft palatal muscle development provides an optimal opportunity to study the initiation, segregation, and migration of muscle anlage formation during craniofacial development.

Clinically, the most common congenital craniofacial malformation is cleft lip with or without cleft palate (*Vyas et al., 2020*). When the soft palate is affected, disruption of the soft palatal shelves separating the oral and nasal cavities co-occurs with defects in soft palatal muscle formation and oropharyngeal function (*Li et al., 2019*; *Carvajal Monroy et al., 2012*). Furthermore, mouse models have demonstrated that loss of key regulators in the perimysial cells, a subset of the palatal mesenchymal connective tissues closely associated with the muscles, ultimately affects soft palatal muscle differentiation and patterning (*Han et al., 2021*; *Sugii et al., 2017*). These findings suggest that CNC-derived perimysial cells possibly contribute to part of the embryonic myogenic niche for craniofacial muscle development at the later stages. Thus, soft palate development can serve as a useful model for studying how perimysial cells regulate late pharyngeal muscle anlage formation through tissue-tissue interactions.

In the present work, we investigated the mechanism that regulates perimysial-to-myogenic communication using the development of the pharyngeal muscle LVP as a model. Using unbiased single-cell RNAseq (scRNAseq) data analysis combined with mouse genetic approaches, we identified TGF-β signaling as a predominant and specific signaling activity that enables the perimysial cells to interact with the adjacent myogenic cells during late pharyngeal muscle development. Furthermore, we identified that a perimysial regulator Creb5 cooperates with TGF-β signaling to co-activate the expression of specific perimysial-to-myogenic signals such as Fgf18 to regulate pharyngeal myogenesis. Taken together, our findings reveal that TGF-β signaling and perimysial-specific regulators may cooperatively define perimysial-to-myogenic signals in regulating pharyngeal myogenesis.

# Results

## Unbiased screening using scRNAseq data analysis identifies TGF-β signaling as predominant and specific signaling for the perimysial cells within the soft palatal mesenchyme

During late pharyngeal muscle development, myogenic cells of the soft palatal muscles, particularly those of the LVP, can be observed segregating from the existing myogenic site associated with middle pharyngeal constrictor to populate the soft palatal shelves and pattern alongside the CNC-derived palatal mesenchymal cells (*Han et al., 2021*; *Li et al., 2019*). In parallel, during this stage, our recent scRNAseq analysis identified that the morphologically homologous palatal mesenchyme is also patterned into heterogeneous cellular domains associated with distinct anatomical locations and gene expression profiles (*Han et al., 2021*). We thus hypothesized that the perimysial population, anatomically adjacent to the myogenic cells, serves as a microenvironment or niche for embryonic pharyngeal muscle development. Furthermore, there must be a specific regulatory mechanism that differentiates the perimysial cells from the rest of the palatal mesenchymal cell populations so that the perimysial cells can provide specific signals to define the myogenic site.

Since signaling pathways have been reported to play critical roles in the patterning and fate determination of craniofacial mesenchymal cells (*Mishina and Snider, 2014*; *Neubüser et al., 1997*; *Vincentz et al., 2016*; *Xu et al., 2019*), we performed an unbiased signaling activity analysis of the cell populations of the soft palate, focusing on identifying key signaling pathways specifically activated in the perimysial cells. Using CellChat's incoming signaling analysis from scRNASeq data of E13.5-E15.5 soft palate, we identified inferred signaling pathway activities of different cell types based on enriched signaling interactions (*Figure 1A–B*, *Figure 1—figure supplement 1*). Notably, the signaling activity of all the palatal mesenchymal cell types was categorized into the same group due to the shared expression of similar signaling pathways (*Figure 1B*), consistent with their common connective tissue identity. Among this group of pathways, Ephrin-Eph (EphA/EphB) (*Benson and Serrano, 2012*; *Xavier et al., 2016*), non-canonical Wnt (ncWnt) (*Reynolds et al., 2019*), IGF (*Marchant et al., 2020*), TGF-β (*Iwata et al., 2011*), FGF (*Nie et al., 2006*), and Hedgehog signaling (HH) pathways (*Everson et al., 2018*) were previously identified to be required for the development of craniofacial mesenchymal tissues including palatal mesenchyme, so we reasoned that they are more likely to be functionally required by the perimysial cell fate determination, too. We thus next analyzed the intensities of activities of these pathways in the perimysial cells and found that the TGF-β signaling activity was inferred to be the highest (*Figure 1C*). Moreover, a known TGF-β signaling downstream target, *Tgfbi* (transforming growth factor-beta-induced protein), was among the top 10 conserved marker genes for the perimysial cells across various developmental stages (*Figure 1D*), suggesting that TGF-β signaling is not only a predominant signaling pathway but also specifically associated with regulating the perimysial cell fate and maintaining a perimysial-derived signal.

To validate the signaling activity analysis result from scRNAseq data, we next evaluated the TGF-β signaling activity *in vivo* during different stages of soft palatal development. Consistent with scRNAseq analysis, TGF-β signaling was present in the palatal shelf mesenchyme primordium during soft palatal development from E12.5 onwards (*Figure 2A–H*) and was most active in the cells surrounding the LVP myogenic cells, anatomically identified as perimysial cells (*Figure 2J–L and N–P*), particularly during the early segregation and separation process at E13.5 (*Figure 2B, F, J, and N*) and E14.5 (*Figure 2C, G, K, and O*). Notably, TGF-β signaling was ubiquitous in the mesenchymal cells close to the pharyngeal wall at E12.5 prior to the arrival of myogenic cells (*Figure 2A, E, I, and M*). This specific activation of TGF-β signaling in the presence of perimysial cells suggested that TGF-β signaling may play a role in patterning perimysial cells to establish a myogenic niche in the specific anatomical location for the LVP. This postulation is supported by our previous work, which showed that loss of TGF-β in *Wnt1-Cre;Tgfbr1^{fl/fl}* mice in the early pre-migratory neural crest cells and their derivatives from E9.5 onward affect proliferation and differentiation of the tongue and other craniofacial muscles (*Chai et al., 2000*; *Han et al., 2014*).

E13.5-E15.5 soft palatal scRNAseq analysis led to the inference that the perimysial cells contain at least two distinct subpopulations with different differentiation statuses, namely earlier perimysial progenitors (*Aldh1a2+*) and more committed perimysial fibroblasts, which express markers other than *Aldh1a2*, such as *Tbx15* (*Han et al., 2021*). We therefore sought to further specify the cellular identity

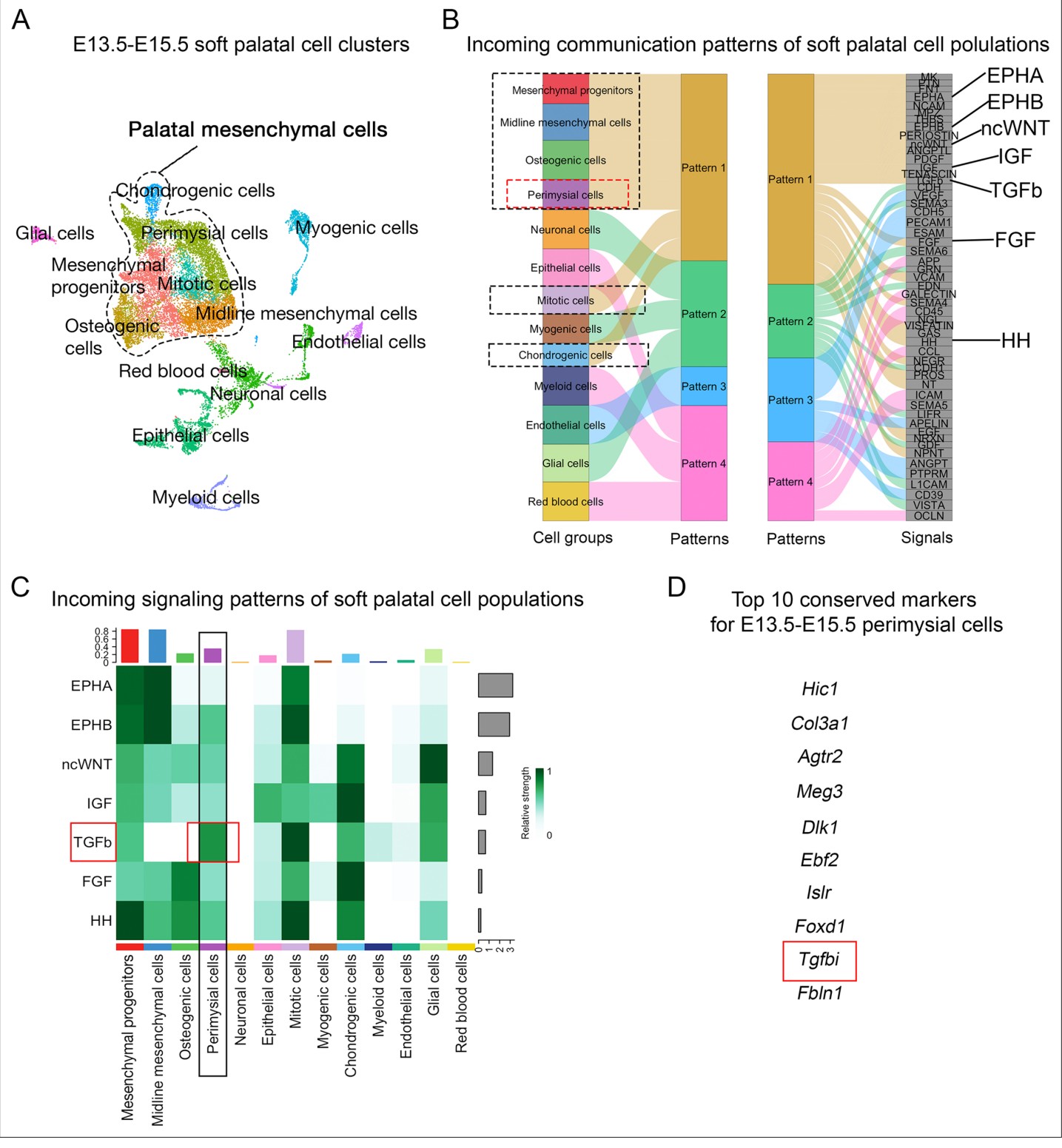

**Figure 1.** CellChat analysis of E13.5-E15.5 soft palatal scRNAseq data predicts TGF-β signaling as a key pathway activated in the perimysial cells. (**A**) UMAP plot of soft palatal cell populations identified by Seurat integration analysis. (**B**) Alluvial plot of the incoming signaling patterns for soft palatal cell populations. The black dotted boxed area indicates the palatal mesenchymal cell populations grouped into the same incoming patterns. The red dotted box highlights the perimysial cells. (**C**) Heatmap of the contribution of the incoming pathways of interest to soft palatal cell populations. The black box highlights signaling activity in the perimysial cells and the red box highlights TGF-β signaling activity in the perimysial cells. (**D**) Top conserved markers for perimysial cells identified through Seurat integration analysis.

*Figure 1 continued on next page*

*Figure 1 continued*

The online version of this article includes the following figure supplement(s) for figure 1:

**Figure supplement 1.** CellChat analysis of scRNAseq data identified incoming signaling pathways for all cell types in the soft palate.

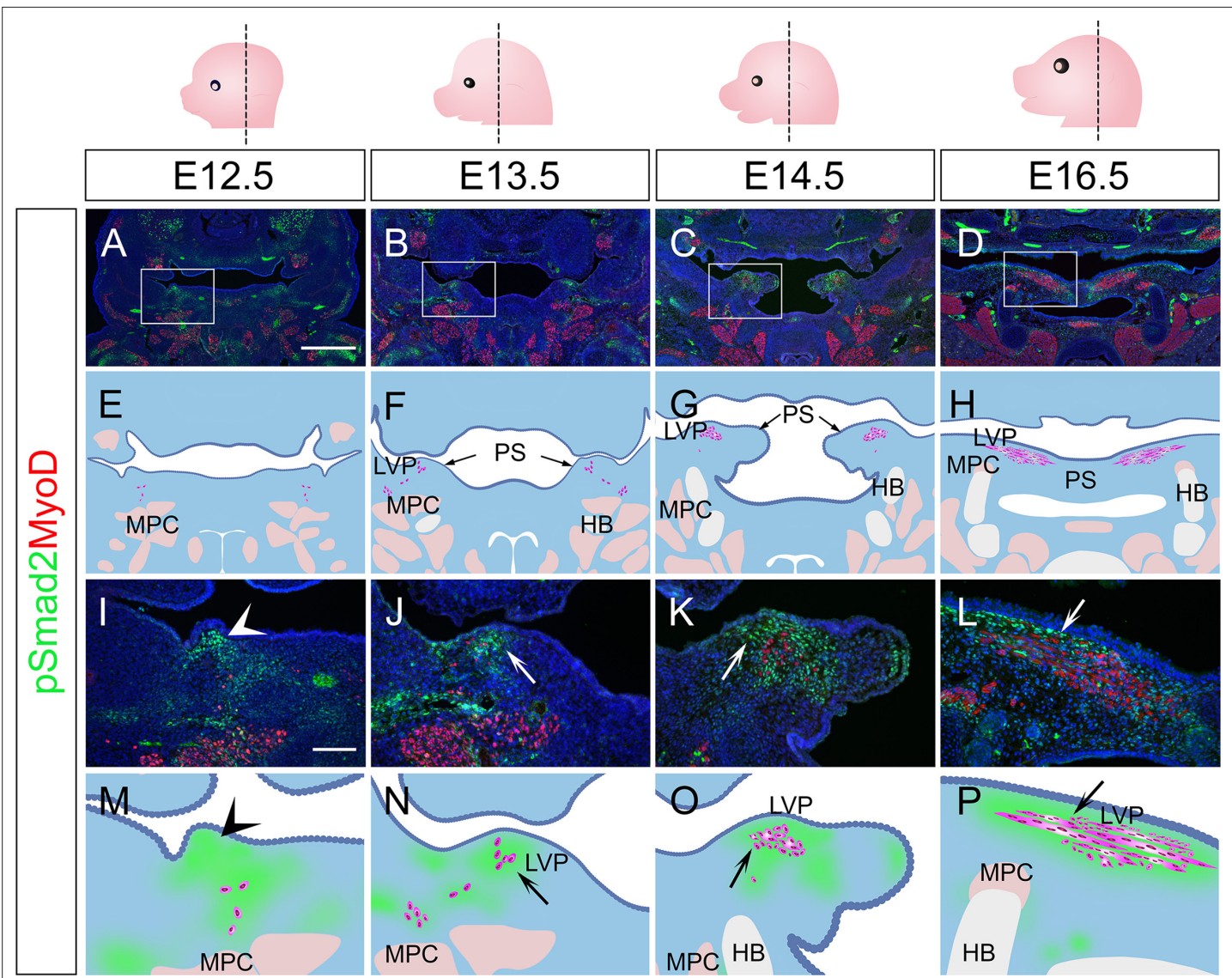

**Figure 2.** TGF-β signaling is activated in perimysial cells surrounding the developing LVP myogenic cells. (**A–P**) Immunofluorescence (**A–D and I–L**) and schematic drawings (**E–H and M–P**) of myogenic marker MyoD (red) and TGF-β signaling readout pSmad2 (green) expression in coronal sections of LVP region at E12.5, E13.5, E14.5, and E16.5. Boxed areas in A, B, C, and D are shown magnified in I, J, K, and L, respectively. Schematic drawings in E-H and M-P correspond to A-D and I-L, respectively. Arrows in F and G indicate the palatal shelves. White and black arrows in J-L and N-P point to the LVP myogenic sites in the palatal shelves. Arrowheads indicate the absence of MyoD+ signal in mouse pharyngeal wall at E12.5 in I and M, and arrows indicate the presence from E13.5 onwards in J-L and N-P. The magenta and salmon colors represent the LVP myogenic cells and the other pharyngeal muscles, respectively. HB, hyoid bone; LVP, levator veli palatini; MPC, middle pharyngeal constrictor; PS, palatal shelves. Top panel schematics depict the orientation and level of the sections. N=3 for all experiments. Scale bars in A and I indicate 500 µm in A-D and 100 µm in I-L, respectively.

The online version of this article includes the following figure supplement(s) for figure 2:

**Figure supplement 1.** *Tbx15+* and *Smoc2+* perimysial cells are present in the region where TGF-β signaling is active.

of the region with TGF-β signaling activity. After subclustering the perimysial cell population from the E13.5-E15.5 soft palatal scRNAseq data, we found these cells could be further divided into 3 clusters (*Figure 2—figure supplement 1A*). Since *Aldh1a2* expression was most enriched in cluster 2 (*Figure 2—figure supplement 1B*), this cluster appeared to be associated with the perimysial progenitor cells, whereas clusters 0 and 1 appeared to be associated with the perimysial fibroblasts. Within the two perimysial fibroblast clusters, the previously reported marker *Tbx15* was expressed by most cells in cluster 0, but not by the majority of cells in cluster 1 (*Figure 2—figure supplement 1C*). Instead, we identified that another perimysial marker, *Smoc2*, labeled both clusters, including the *Tbx15*-negative cells (*Figure 2—figure supplement 1D*). We confirmed that *Smoc2* expression was specific to the perimysial populations among all the palatal mesenchymal populations (*Figure 2—figure supplement 1E*). We next mapped the expression of these markers *in vivo* using E14.5 palatal mesenchyme, at which stage distinct domains for different cell types have already been established. Consistent with the scRNAseq data analysis result, *Aldh1a2+* perimysial progenitors and *Tbx15+* or *Smoc2+* perimysial fibroblasts were found to be associated with different cellular identities *in vivo*. The *Aldh1a2*+population was located at the base of the palatal shelf and closer to the middle pharyngeal constrictor myogenic sites outside the palatal shelf; it had little overlap with the pSmad2+ area (*Figure 2—figure supplement 1G and J*). Both *Tbx15* and *Smoc2* were expressed in the palatal shelf, but they appeared to overlap with pSmad2+ cells in certain locations. *Tbx15* expression was more restricted to the center of the palatal shelf, thus overlapping with the center region of the pSmad2+ cells (*Figure 2—figure supplement 1H and K*). *Smoc2* was more extensively expressed and overlapped at the periphery of the pSmad2 expression area, almost complementary to the *Tbx15+* domain (*Figure 2—figure supplement 1I and L*). This overlap between TGF-β signaling activity and the *Tbx15+* and *Smoc2+* populations suggested that TGF-β signaling may be a more specific regulator for the later stage of perimysial cells (perimysial fibroblasts), not the perimysial progenitors.

## Loss of TGF-β leads to defects in soft palatal shelf mesenchyme and LVP in *Osr2^{Cre};Tgfbr1^{fl/fl}* mice

To test whether this specific TGF-β signaling in the perimysial cells in later developmental stages also plays any functional role in regulating the fate of myogenic cells, we utilized *Osr2^{Cre}*, which affects the palatal mesenchyme at a later stage than *Wnt1-Cre* (*Chen et al., 2009*). Using *Osr2^{Cre};Rosa26^{LSL-tdTomato}* mice, we confirmed that *Osr2^{Cre}* can specifically label the palatal mesenchyme without affecting the myogenic cells during soft palatal muscle development (*Figure 3—figure supplement 1A–D*). Thus, we generated *Osr2^{Cre};Tgfbr1^{fl/fl}* mice, in which TGF-β signaling is ablated in the palatal mesenchyme due to the loss of TGF-β Type 1 receptor *Alk5* (*Tgfbr1*). Loss of TGF-β signaling in the palatal mesenchyme, including the perimysial cells, led to morphologically deformed palatal shelves and cleft palate in the LVP region in E18.5 *Osr2^{Cre};Tgfbr1^{fl/fl}* mice (*Figure 3B and D*), compared with the intact control palatal shelves (*Figure 3A and C*), indicating intrinsic defects in the palatal mesenchymal cells. In addition, while abundant MHC+ muscle fibers were present in the control palatal shelves (*Figure 3E, I, G, and K*), in the defective palatal shelves of *Osr2^{Cre};Tgfbr1^{fl/fl}* mice, LVP formation was so severely affected that no muscle fibers could be detected at this stage (*Figure 3F, J, H, and L*). This confirmed that TGF-β signaling is indispensable for the perimysial-derived signaling for myogenesis.

We next evaluated the progression of the palate mesenchymal and myogenic cell defects in the *Osr2^{Cre};Tgfbr1^{fl/fl}* mice. We found that the palatal shelves of *Osr2^{Cre};Tgfbr1^{fl/fl}* mice started to appear morphologically different from those of the controls at around E14.5, suggesting that defects in the palatal mesenchymal cells start at this stage (*Figure 3—figure supplement 2A–H*). However, the myogenic defects could be observed as early as E13.5 in the *Osr2^{Cre};Tgfbr1^{fl/fl}* mice while the palatal shelves remained comparable between *Osr2^{Cre};Tgfbr1^{fl/fl}* and control mice (*Figure 3—figure supplement 3*). As myogenic cells began to enter the palatal shelves at E13.5, fewer myogenic cells were present in the *Osr2^{Cre};Tgfbr1^{fl/fl}* mice compared to the control (*Figure 3—figure supplement 3A, E, and I*), suggesting a defect affecting the migration of the myogenic cells. At E14.0, this reduced population of myogenic cells also showed significantly less proliferative activity in the *Osr2^{Cre};Tgfbr1^{fl/fl}* mice (*Figure 3—figure supplement 3B–C, F–G, and J–K*). No apoptosis was observed in either the control or *Osr2^{Cre};Tgfbr1^{fl/fl}* myogenic cells at E14.0 (*Figure 3—figure supplement 3D and H*). At E14.5, myogenic cells were still present in the *Osr2^{Cre};Tgfbr1^{fl/fl}* mice, although fewer in number than in the control (*Figure 3—figure supplement 2A–D and I*). From E14.5 onward, the palatal mesenchymal

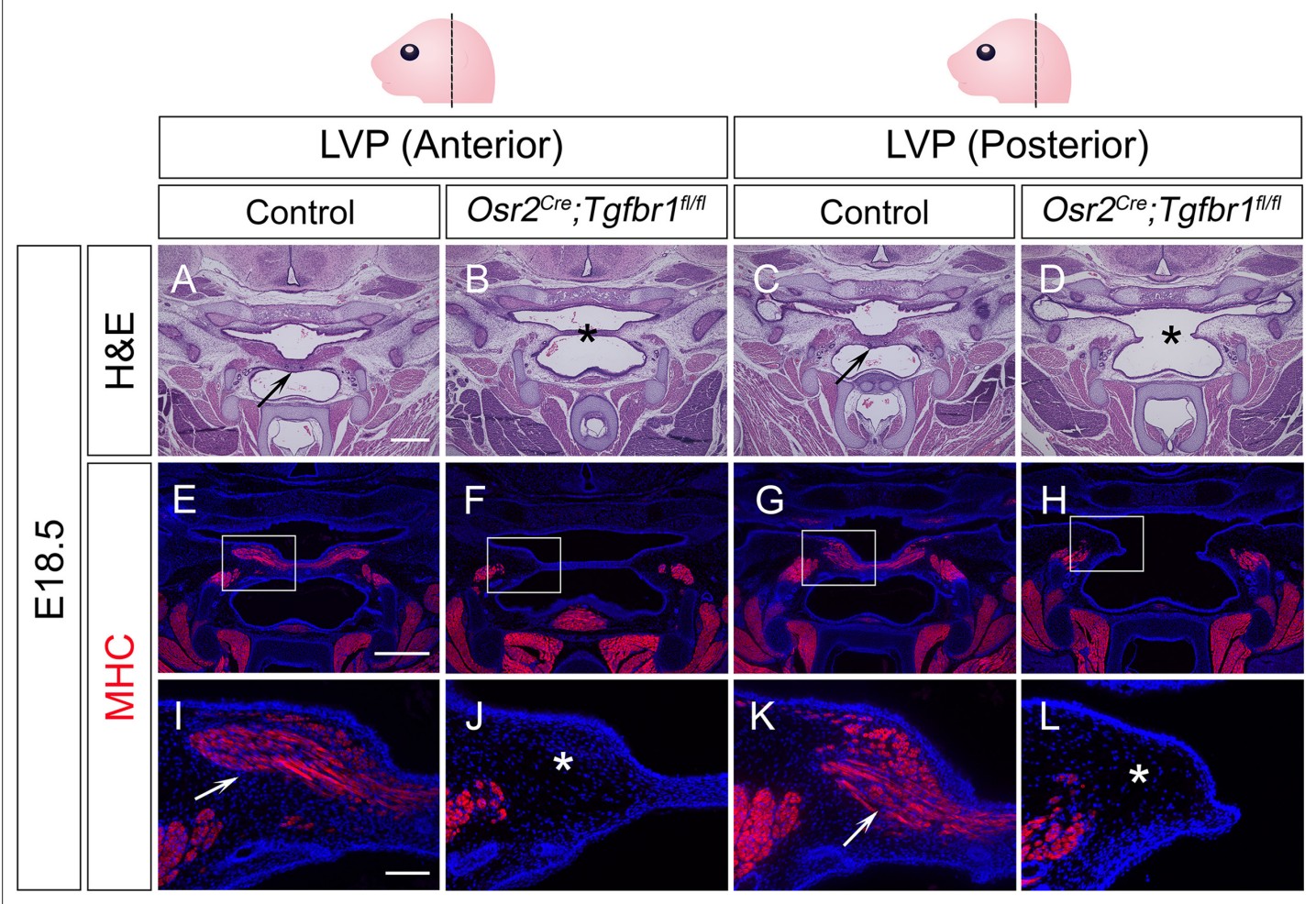

**Figure 3.** *Osr2^Cre^;Tgfbr1^fl/fl^* mice exhibit palatal shelf and LVP defects at E18.5. (**A–D**) H&E staining of coronal sections at LVP region from control and *Osr2^Cre^;Tgfbr1^fl/fl^* mice at E18.5. Arrows indicate the normal palatal shelf in A and C; asterisks indicate the deformed and cleft palatal shelf in B and D, respectively. N=4. (**E–L**) Immunofluorescence of MHC (red) in coronal sections at LVP region from control and *Osr2^Cre^;Tgfbr1^fl/fl^* mice at E18.5. Boxed areas in E, F, G, and H are enlarged in I, J, K, and L, respectively. Arrows indicate the presence of muscle fibers in I and K; asterisks indicate their absence in J and L. N=3. Top panel schematics depict the orientation and level of the sections. *Tgbfr1^fl/fl^* or *Tgbfr1^fl/+^* littermates were used as controls for *Osr2^Cre^;Tgfbr1^fl/fl^* mice. Scale bars in A and E indicate 500 µm in A-D and E-H, respectively; the scale bar in I indicates 100 µm in I-L.

The online version of this article includes the following source data and figure supplement(s) for figure 3:

**Figure supplement 1.** *Osr2^Cre^* targets palatal mesenchymal cells surrounding soft palatal myogenic cells.

**Figure supplement 2.** *Osr2^Cre^;Tgfbr1^fl/fl^* mice exhibit palatal shelf and myogenic defects in the LVP region from E14.5 onwards.

**Figure supplement 2—source data 1.** Source data for *Figure 3—figure supplement 2I*.

**Figure supplement 3.** Myogenic defects occur in *Osr2^Cre^;Tgfbr1^fl/fl^* mice prior to E14.5.

**Figure supplement 3—source data 1.** Source data for *Figure 3—figure supplement 3I*.

**Figure supplement 3—source data 2.** Source data for *Figure 3—figure supplement 3J*.

**Figure supplement 3—source data 3.** Source data for *Figure 3—figure supplement 3K*.

defects became more exacerbated over time, and the myogenic defect also increased in severity over time as myogenic progenitor cells became almost undetectable from around E16.5 (*Figure 3—figure supplement 2E–H*). We also confirmed that the TGF-β signaling activity (pSmad2) was reduced in the perimysial cells adjacent to the LVP in the *Osr2^Cre^;Tgfbr1^fl/fl^* mice (*Figure 3—figure supplement 2J–M*). Taken together, our results indicated that loss of TGF-β signaling activity in the palatal mesenchyme leads to migration and proliferation defects of LVP myogenic cells in *Osr2^Cre^;Tgfbr1^fl/fl^* mice through cell-cell interaction.

## TGF-β signaling is required for perimysial fibroblasts in regulating pharyngeal muscle development

To precisely identify the molecular changes of the perimysial population in the *Osr2^Cre^;Tgfbr1^fl/fl^* mice, we performed scRNAseq analysis to compare cell-type-specific gene expression profiles of cell populations in E14.5 *Osr2^Cre^;Tgfbr1^fl/fl^* and control soft palates. Using integration analysis based on shared variance (*Figure 4—figure supplement 1A*), we identified similar clusters in the soft palates of control and *Osr2^Cre^;Tgfbr1^fl/fl^* mice at this stage. We first distinguished the palatal mesenchymal cells (*Runx2+/Twist1+*) from other non-mesenchymal cell types in the soft palate (*Figure 4—figure supplement 1B–L*). Then, using more detailed markers we recently established from E13.5-E15.5 soft palatal scRNAseq analysis (*Han et al., 2021*), we further distinguished the perimysial population (*Tbx15+/Hic1+/Aldh1a2+*) from the remaining palatal mesenchymal populations of the E14.5 *Osr2^Cre^;Tgfbr1^fl/fl^* and control mice (*Figure 4—figure supplement 1M–N*).

To more specifically evaluate the changes in the perimysial cells following the loss of TGF-β signaling, we subsetted the perimysial population from the mesenchymal clusters of the integrated *Osr2^Cre^;Tgfbr1^fl/fl^* and control soft palate scRNAseq data at E14.5 (*Figure 4A*). We first compared the expression of perimysial progenitor and perimysial fibroblast markers between the *Osr2^Cre^;Tgfbr1^fl/fl^* and control perimysial cells at E14.5 (*Figure 4B–D*), to identify whether one or both subpopulations of the perimysial cells were affected. Consistent with the overlapping of perimysial fibroblast populations in the region of active TGF-β signaling, in our scRNAseq data we found reduced expression of perimysial fibroblast markers *Tbx15* and *Smoc2* in the perimysial population in E14.5 *Osr2^Cre^;Tgfbr1^fl/fl^* soft palates relative to controls (*Figure 4B and C*), while the *Aldh1a2+* perimysial progenitor subpopulation was not affected (*Figure 4D*). More specifically, *in vivo* expression patterns further showed that *Tbx15+* cells were located predominantly in between the LVP myogenic cells, while *Smoc2* expression was present mostly around the periphery of the myogenic site (*Figure 4F and H*). Consistent with the scRNAseq analysis, expression of *Tbx15* and *Smoc2* was affected in perimysial cells in *Osr2^Cre^;Tgfbr1^fl/fl^* mice at these respective sites (*Figure 4G and I*). Also consistent with the scRNAseq analysis, the *Aldh1a2+* cells located mainly adjacent to the middle pharyngeal constrictor in the control were not affected in the *Osr2^Cre^;Tgfbr1^fl/fl^* mice (*Figure 4J and K*). While our recent study showed that Runx2 is a regulator of *Aldh1a2+* early perimysial progenitors (*Han et al., 2021*), here we identified TGF-β signaling as the main regulator for the fate determination of both populations of the more committed perimysial fibroblasts, suggesting a specific role of TGF-β signaling in late stage-pharyngeal muscle development.

Since TGF-β signaling has been reported to regulate proliferation of the hard palate mesenchyme (*Ito et al., 2003*), we further investigated whether the loss of *Tbx15* and *Smoc2* at E14.5 in the *Osr2^Cre^;Tgfbr1^fl/fl^* mice was due to the failure of palatal mesenchymal cells to survive, differentiate, and/or proliferate. To evaluate the activation of these two markers during development, we analyzed their expression patterns at E13.5-E14.0 in control and *Osr2^Cre^;Tgfbr1^fl/fl^* mice. We found that *Tbx15* expression was not activated at early stages: *Tbx15* was not detectable E13.5, and only starting to be activated at a low level by a few cells at E14.0 in both and *Osr2^Cre^;Tgfbr1^fl/fl^* mice (*Figure 4—figure supplement 2A–D*). This low level of *Tbx15* increased dramatically at E14.5 in the control (*Figure 4F*) but failed to increase in a similar manner in *Osr2^Cre^;Tgfbr1^fl/fl^* mice (*Figure 4G*), suggesting that the reduced *Tbx15* expression at E14.5 was due to a failure of activating the differentiation of *Tbx15+* cells. In parallel, we found that *Smoc2* was expressed extensively in the palatal mesenchyme at E13.5 at comparable levels in control and *Osr2^Cre^;Tgfbr1^fl/fl^* mice, and its expression started to reduce in the *Osr2^Cre^;Tgfbr1^fl/fl^* mice at E14.0 (*Figure 4—figure supplement 2E–J*). Since the proliferation rate in the palatal shelves and the *Smoc2+* cells were not affected at E13.5 (*Figure 4—figure supplement 2K–P*), and very few cells were apoptotic in the palatal mesenchyme (less than 1%) in both the control and *Osr2^Cre^;Tgfbr1^fl/fl^* mice at this stage (*Figure 4—figure supplement 2Q–R*), we inferred that the loss of *Smoc2* was likely due to the differentiation defect of the *Smoc2* populations, too. Thus, we concluded that TGF-β signaling is required for the fate determination of both *Tbx15+* and *Smoc2+* perimysial fibroblasts during early palatal mesenchyme development.

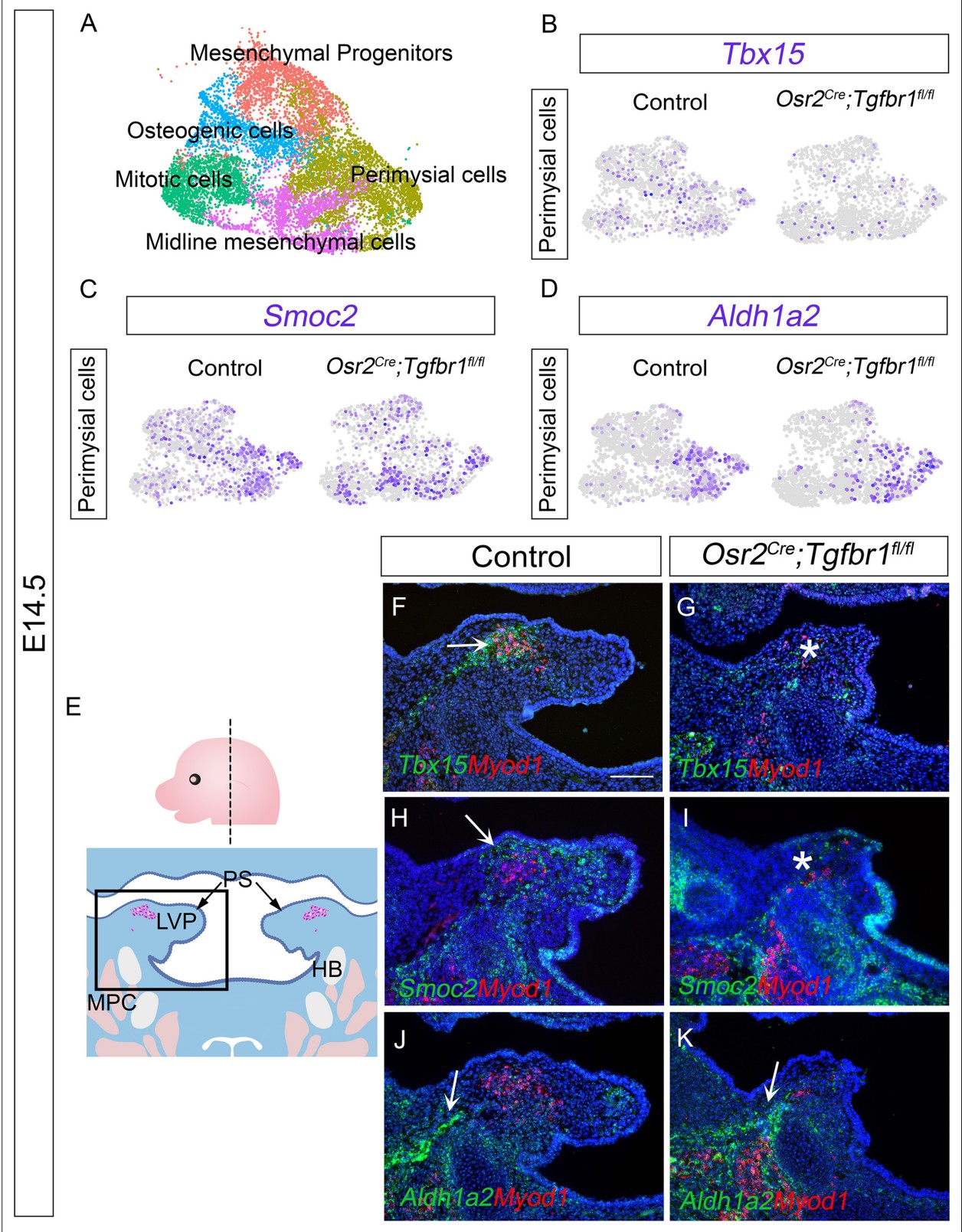

**Figure 4.** The *Tbx15+* and *Smoc2+* subpopulations are more affected than the *Aldh1a2+* subpopulation in *Osr2^Cre^;Tgfbr1^fl/fl^* palatal mesenchyme at E14.5. (**A**) UMAP plot of palatal mesenchymal cell clusters from integrated scRNAseq analysis of control and *Osr2^Cre^;Tgfbr1^fl/fl^* soft palates at E14.5. (**B–D**) Feature plot view of expression patterns of perimysial fibroblast markers *Tbx15* (**B**) and *Smoc2* (**C**) and perimysial progenitor marker gene *Aldh1a2* (**D**) in perimysial cells from integrated control and *Osr2^Cre^;Tgfbr1^fl/fl^* palatal mesenchymal cell clusters at E14.5. (**E**) Schematic drawing of orientation and

*Figure 4 continued*

level of the sections (top panel) and coronal sections of LVP region at E14.5 (bottom panel). Boxed area in E indicates the region of F-K. Magenta and salmon colors represent the LVP myogenic cells and the other pharyngeal muscles, respectively. HB, hyoid bone; LVP, levator veli palatini; MPC, middle pharyngeal constrictor; PS, palatal shelves. (**F–K**) RNAScope *in situ* hybridization for *Tbx15* (green), *Smoc2* (green), or *Aldh1a2* (green) colocalization with myogenic markers *Myod1* (red) in coronal sections of LVP region of control and *Osr2^Cre^;Tgfbr1^fl/fl^* mice at E14.5. Arrows in F, H, J, and K indicate *Tbx15*, *Smoc2*, or *Aldh1a2* expression. Asterisks in G and I show reduced *Tbx15* or *Smoc2* expression. N=3. *Tgbfr1^fl/fl^* or *Tgbfr1^fl/+^* littermates were used as controls for *Osr2^Cre^;Tgfbr1^fl/fl^* mice. The scale bar in F indicates 100 µm in F-K.

The online version of this article includes the following source data and figure supplement(s) for figure 4:

**Figure supplement 1.** scRNAseq analysis identified distinct populations of palatal mesenchymal cells in E14.5 *Osr2^Cre^;Tgfbr1^fl/fl^* and control (*Tgbfr1^fl/fl^* or *Tgfbr1^fl/+^*) soft palates.

**Figure supplement 2.** Expression of perimysial markers *Tbx15* and *Smoc2* in E13.5-E14.0 *Osr2^Cre^;Tgfbr1^fl/fl^* and control soft palates.

**Figure supplement 2—source data 1.** Source data for *Figure 4—figure supplement 2I*.

**Figure supplement 2—source data 2.** Source data for *Figure 4—figure supplement 2J*.

**Figure supplement 2—source data 3.** Source data for *Figure 4—figure supplement 2O*.

**Figure supplement 2—source data 4.** Source data for *Figure 4—figure supplement 2P*.

## CellChat analysis of soft palatal scRNAseq data identifies putative perimysial-to-myogenic signaling molecules downstream of TGF-β signaling

Since the perimysial fibroblast population is anatomically adjacent to the myogenic cells, they are most likely to be part of a microenvironment that supports myogenesis through signaling communication. To systemically identify potential signaling interactions between these two cell populations, we examined the E13.5-E15.5 integrated soft palatal scRNAseq data with the outgoing signaling analysis of the CellChat package (*Han et al., 2021*; *Jin et al., 2021*) to infer enriched intercellular signaling interactions at the single-cell level. In particular, we focused our analysis on signals sent from perimysial fibroblasts and received by myogenic cells. In doing so, we detected several enriched interactions including those between perimysial-derived signaling molecules in ncWnt, FGF, Notch, and BMP signaling pathways (Wnt5a, Fgf18, Dlk1, and Bmp4) and corresponding receptors expressed in the myogenic cells (Fdz4, Fgfr1/4, Notch3, and Bmpr1a/Acvr2b, respectively) (*Figure 5A*). We also identified other interactions that may be associated with cell behavior through modulating extra-cellular matrices, such as Thrombospondin (Thbs3-Sdc1), Pleiotrophin (Ptn-Ncl), Nectins (Nectin3-Nectin1), Midkine (Mdk-Ncl), and Laminins (Lama4/b1/c1-Dag1) (*Figure 5A*).

To identify perimysial-to-myogenic interactions that are functionally required for regulating myogenic fate, we decided to narrow our selection to the signaling molecules more specifically expressed by the perimysial fibroblasts. By checking the expression patterns of these signaling molecules in E13.5-E15.5 soft palatal scRNAseq data, we found that *Lama4*, *Fgf18*, and *Dlk1* are the top three molecules more enriched in the perimysial fibroblasts than in other cells (*Figure 5B*). This finding suggests they play a more specific role in regulating myogenesis. Among these three molecules, expression of *Fgf18* and *Lama4* was found to be reduced in the *Osr2^Cre^;Tgfbr1^fl/fl^* perimysial cells, whereas *Dlk1* expression appeared unaffected (*Figure 5C–F*). This suggests that TGF-β signaling could also regulate the expression of perimysial-derived *Lama4* and *Fgf18* during muscle development. *In vivo* expression of these two genes further showed that *Fgf18* expression is more restricted to the region of perimysial fibroblasts (*Figure 5H*) than that of *Lama4* (*Figure 5I*), similar to the TGF-β signaling activity. Thus, we identified *Fgf18* as the most likely perimysial-to-myogenic signal regulated by TGF-β signaling to support myogenesis.

To better understand the role of perimysial-myogenic signals during myogenesis, we further investigated the *in vivo* expression of *Fgf18* in perimysial cells during LVP development in control and *Osr2^Cre^;Tgfbr1^fl/fl^* mice. We found consistently enriched expression of *Fgf18* in the location of perimysial fibroblasts most adjacent to the myogenic cells between E13.5-E16.5 (*Figure 5J–K, N–O, and R–S*). Consistent with the scRNAseq analysis, the expression of *Fgf18* in similar locations in the *Osr2^Cre^;Tgfbr1^fl/fl^* mice was reduced compared to the ones in controls (*Figure 5L–M, P–Q, and T–W*). This reduction of *Fgf18* expression occurred as early as E13.5, during which the defect was apparent in the myogenic cells but not in the palatal mesenchyme, suggesting TGF-β signaling regulates *Fgf18* expression prior to changes to the perimysial cell fates (*Figure 5J–M* and *Figure 4—figure supplement 2A, C, E, and*

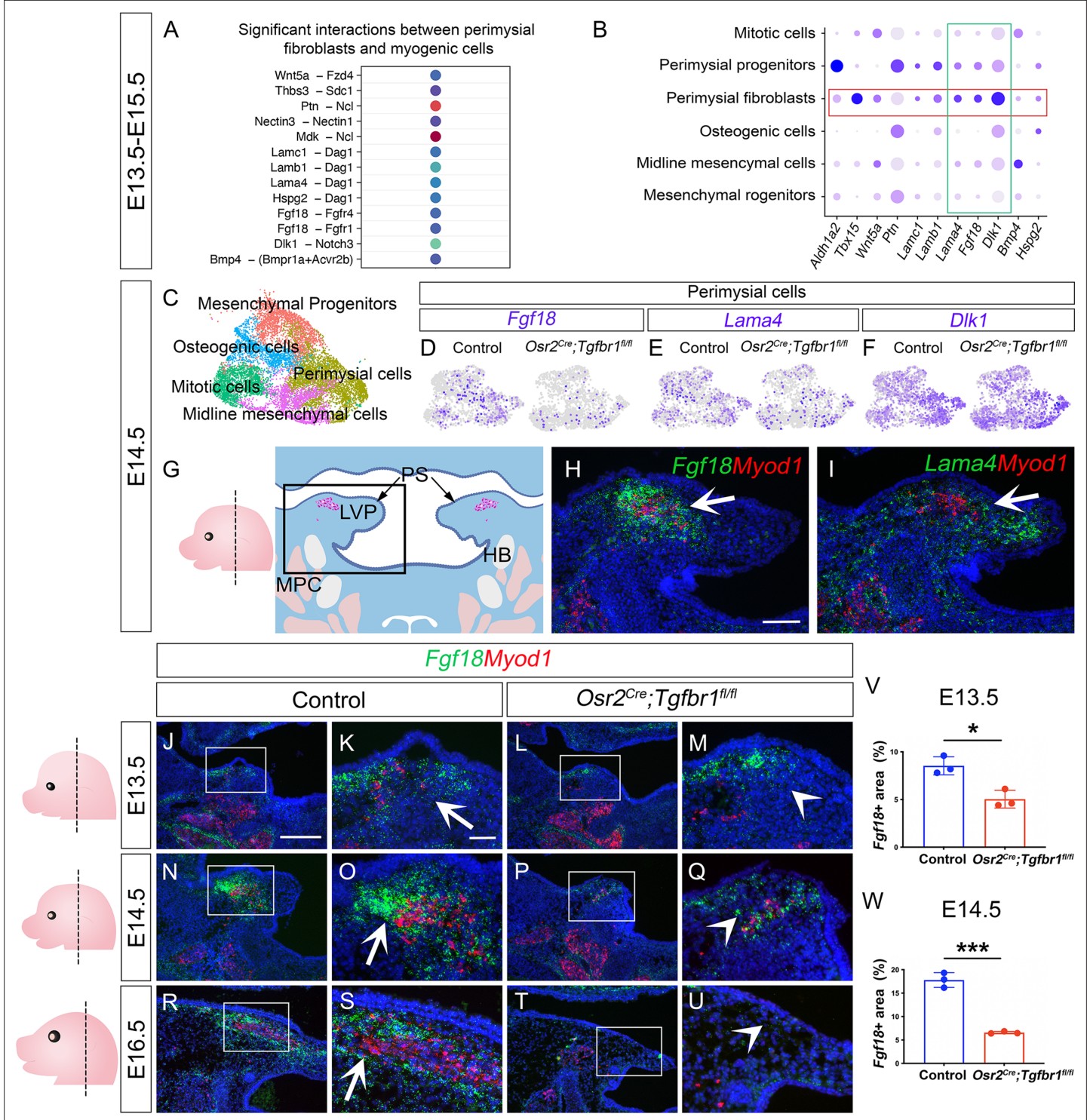

**Figure 5.** CellChat analysis of soft palatal scRNAseq data identifies putative perimysial-to-myogenic signaling molecules downregulated in *Osr2^Cre^;Tgfbr1^fl/fl^* perimysial fibroblasts. (**A**) CellChat analysis of scRNAseq integration data of E13.5-E15.5 soft palatal tissue showing significant interactions between perimysial fibroblasts and myogenic cells. (**B**) Dot plot of expression patterns of perimysial fibroblast-derived signals in individual cellular clusters of integration scRNAseq from E13.5-E15.5 soft palatal tissue. Note that highlighted signals, *Lama4*, *Fgf18*, and *Dlk1*, are more enriched in the perimysial fibroblasts. (**C**) UMAP plot of palatal mesenchymal cell clusters from integrated scRNAseq analysis of control and *Osr2^Cre^;Tgfbr1^fl/fl^* soft palates at E14.5. (**D–F**) Feature plot view of *Fgf18* (**D**), *Lama4* (**E**), or *Dlk1* (**F**) expression in perimysial cells from the integrated control and *Osr2^Cre^;Tgfbr1^fl/fl^* palatal mesenchymal cell clusters at E14.5. (**G**) Schematic drawing of orientation and level of the sections (left panel) and coronal section of the LVP region (right panel) at E14.5. Boxed area in G indicates the region shown in H-I. The magenta and salmon colors represent the LVP

*Figure 5 continued on next page*

*Figure 5 continued*

myogenic cells and the other pharyngeal muscles, respectively. HB, hyoid bone; LVP, levator veli palatini; MPC, middle pharyngeal constrictor; PS, palatal shelves. (**H–I**) RNAScope *in situ* hybridization for *Myod1* (red) and *Fgf18* (green) (**H**) or *Lama4* (green) (**I**) in the coronal section of the LVP region at E14.5. White arrows in H and I point to a positive signal. (**J–U**) RNAScope *in situ* hybridization for *Myod1* (red) and *Fgf18* (green) in the coronal section of the LVP region at E13.5 (**J–M**), E14.5 (**N–Q**), and E16.5 (**R–U**) in control and *Osr2^{Cre}*;*Tgfbr1^{fl/fl}* mice. Boxed areas in J, L, N, P, R, and T are enlarged in K, M, O, Q, S, and U, respectively. Arrows in K, O, and S point to positive signal; arrowheads in M, Q, and U indicate lack of signal. Left panel schematics depict the orientation and level of the sections. (**V–W**) Quantification of the percentage of *Fgf18* expressing area out of the entire the palatal shelf region for control and *Osr2^{Cre}*;*Tgfbr1^{fl/fl}* mice at E13.5 (**V**) and E14.5 (**W**). *, p≤0.05; ***, p≤0.001. Statistical significance was assessed by unpaired t-test with two-tailed calculations. Data is presented as mean ± SEM. N=3 for all experiments. *Tgbfr1^{fl/fl}* or *Tgbfr1^{fl/+}* littermates were used as controls for *Osr2^{Cre}*;*Tgfbr1^{fl/fl}* mice. The scale bar in H indicates 100 μm in H and I; the scale bar in J indicates 250 μm in J, L, N, P, R, and T; the scale bar in K indicates 50 μm for the rest of the figures.

The online version of this article includes the following source data and figure supplement(s) for figure 5:

Source data 1. Source data for *Figure 5U*.

Source data 2. Source data for *Figure 5V*.

Figure supplement 1. Fgf18 receptor *Fgfr4* is predominantly expressed by myogenic cells in the LVP regions.

*G*). Notably, the reduction of the myogenic cells was correlated with the loss of *Fgf18* expression. In the *Osr2^{Cre}*;*Tgfbr1^{fl/fl}* soft palate at E14.5, a reduced number of *Myod1+* myogenic cells was detectable in the location of the remnant *Fgf18* expression compared with the presence of myogenic cells in control (*Figure 5O and Q*). Further, at E16.5, the *Myod1* expression became undetectable as *Fgf18* expression disappeared compared with the abundant presence of both signals in the control (*Figure 5R–U*). This strong correlation between the perimysial-fibroblast-derived *Fgf18* and *Myod1+* myogenic cells further supported the potential role of Fgf18 in myogenesis. We therefore next evaluated the expression of FGF receptors in the MyoD+ myogenic cells to test whether myogenic cells could receive an Fgf18 signal. Consistent with CellChat predictions, perimysial fibroblast-derived Fgf18 signals were more likely to communicate with Fgfr1 and/or Fgfr4 in the myogenic cells, which did not express *Fgfr2* or *Fgfr3* (*Figure 5—figure supplement 1B, C, F, and G*). Though *Fgfr1* is expressed extensively in the palatal mesenchymal cells and weakly in myogenic cells (*Figure 5—figure supplement 1A and E*), *Fgfr4* showed more specific and stronger expression in the myogenic cells (*Figure 5—figure supplement 1D and H*). To further explore which myogenic subpopulation may respond to the Fgf18 signals, we examined the expression pattern of *Fgfr4* in the myogenic cells from the E13.5-E15.5 soft palatal scRNAseq data. We found that *Fgfr4* expression was mostly enriched in the myogenic cells expressing early myogenic markers *Myf5* and *Pax7* (*Figure 5—figure supplement 1I–K*), but not in the more differentiated cells (*Myog+/Myl1+*) (*Figure 5—figure supplement 1L and M*). Interestingly, these myogenic progenitor cells also expressed the proliferative marker *Mki67* (*Figure 5—figure supplement 1N*), consistent with the proliferative status of myogenic progenitors. Consistent with the scRNAseq analysis, *Myf5+* cells co-expressed *Fgfr4* in the palatal shelf sections *in vivo* (*Figure 5—figure supplement 1O and P*). Thus, perimysial-fibroblast-derived Fgf18 may signal to Fgfr4 in the myogenic progenitors to orchestrate myogenic cell fate and promote their proliferation.

## Smad2/3 and Creb5 cooperatively regulate Fgf18 signaling for pharyngeal muscle development

To investigate how TGF-β signaling serves as a key regulator of perimysial-to-myogenic communication, particularly Fgf18 signaling, we first explored whether TGF-β signaling mediators Smad2/3 can directly bind to the promoter region of *Fgf18*. Using JASPAR TF binding sites (TFBSs) predictions incorporated in the UCSC Genome Browser (*Castro-Mondragon et al., 2022*), we identified that several putative TFBSs for Smad2/3 are present in the promoter region 2 kb upstream of the transcription start site (TSS) (*Figure 6A*) and focused on the binding motif at chr11:33098353–33098362, which had the highest predicted binding score, since it is most likely to be the binding site (*Figure 6A*). Using a Cut and Run assay for Smad2/3, we identified significantly greater binding Smad2/3 to DNA fragments around this binding motif when compared to IgG control (*Figure 6B*), suggesting direct binding of Smad2/3 to this predicted TFBS in the *Fgf18* promoter region.

To further test the functional requirment of this TFBS, we repressed the transcription initiation of this TFBS using a CRISPRi system that sterically prevents the association between DNA motifs and transcription factors when the gRNA is targeting the promoter region (*Larson et al., 2013*), which

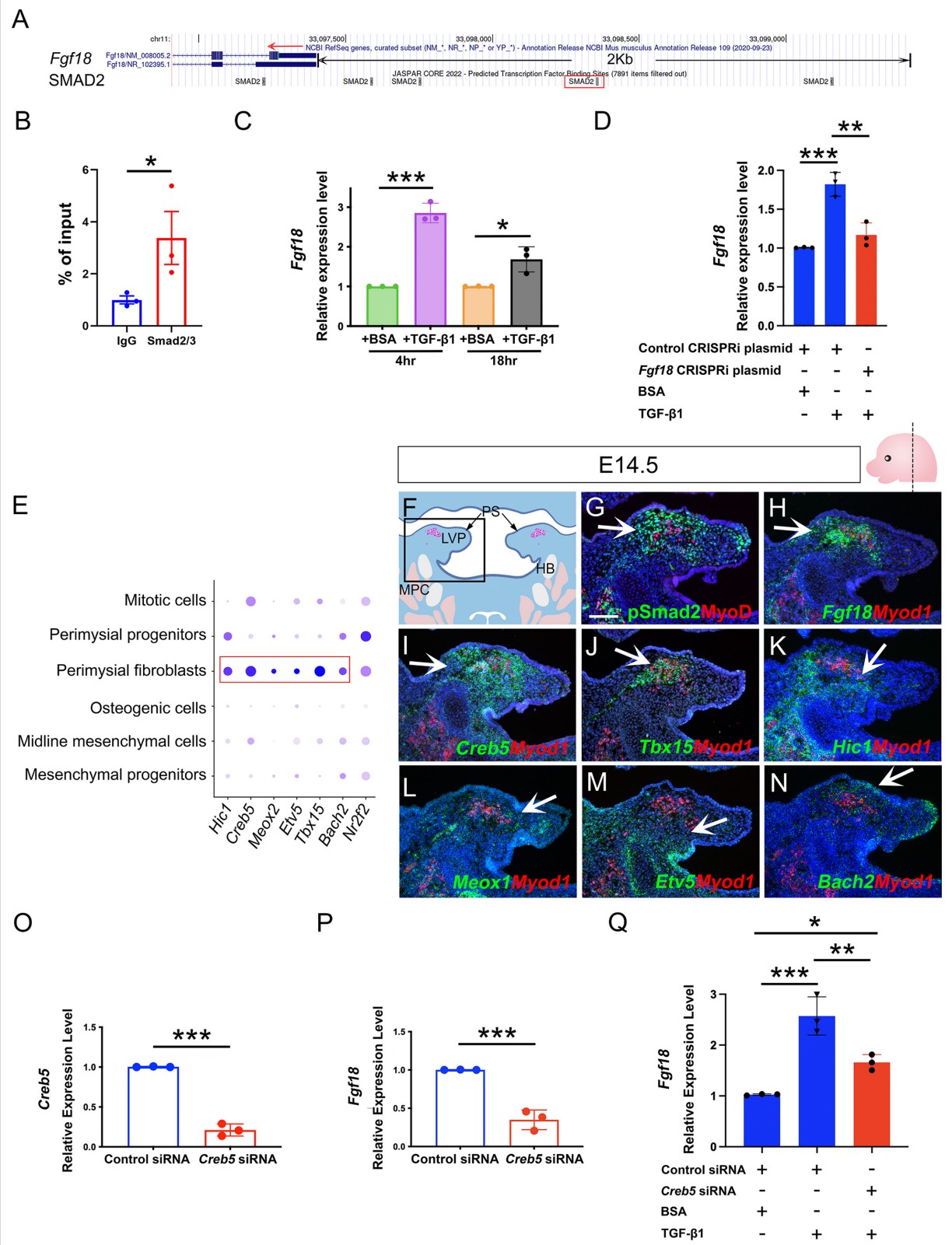

**Figure 6.** Perimysial fibroblast-specific regulon Creb5, identified by SCENIC analysis, cooperates with TGF-β signaling to regulate *Fgf18*. (**A**) UCSC binding prediction of SMAD2 binding motif to the promoter of *Fgf18* gene. The boxed area indicates the binding site with the highest score. (**B**) Cut and Run analysis shows significantly more enriched Smad2/3 binding to the promoter region of the *Fgf18* gene close to the predicted binding site than that of IgG in the soft palatal tissue of the control mice at E14.5. *, p-value ≤0.05. Statistical significance was assessed by unpaired t-test with two-tailed

*Figure 6 continued*

calculations. Data is presented as mean ± SEM. (**C**) qPCR analysis of *Fgf18* expression in E14.5 soft palatal mesenchymal cell culture after treatment with 5 ng/ml TGF-β1 or BSA compared after 4 or 18 hr. Note the increase of *Fgf18* at 4 hr is higher than at 18 hr. *, p≤0.05; ***; p≤0.001. Statistical significance was assessed by unpaired t-test with two-tailed calculations. Data is presented as mean ± SEM. (**D**) qPCR analysis of *Fgf18* expression of E14.5 soft palatal mesenchymal cell culture transfected with *Fgf18* or Control CRISPRi plasmid followed by treatment with 5 ng/ml TGF-β1 or BSA for 4 hr. '+' or '-' under the plots indicates the presence or absence of the indicated treatment. **, p≤0.01; ***, p≤0.001. Statistical significance was assessed by ANOVA. Data is presented as mean ± SEM. (**E**) Dotplot of perimysial fibroblast regulon expression pattern in individual cellular clusters of integrated scRNAseq from E13.5-E15.5 soft palatal tissue. Highlighted regulons are more enriched in the perimysial fibroblasts. (**F**) Schematic drawing of coronal section of the LVP region at E14.5. Boxed area in E indicates the region shown in G-N. The magenta and salmon colors represent the LVP myogenic cells and the other pharyngeal muscles, respectively. HB, hyoid bone; LVP, levator veli palatini; MPC, middle pharyngeal constrictor; PS, palatal shelves. (**G**) Immunofluorescence of MyoD (red) and pSMAD2 (green) in the coronal section of the LVP region at E14.5. (**H–N**) RNAScope *in situ* hybridization for *Myod1* (red) and *Fgf18* (green) (**H**), *Creb5* (green) (**I**), *Tbx15* (green) (**J**), *Hic1* (green) (**K**), *Meox1* (green) (**L**), *Etv5* (green) (**M**), or *Bach2* (green) (**N**) in coronal sections of the LVP region at E14.5. Arrows indicate positive signals. (**O–P**) qPCR analysis of *Creb5* (**O**) and *Fgf18* expression (**P**) after *Creb5* siRNA treatment on E14.5 soft palatal mesenchymal cell culture compared with the control siRNA. ***, p≤0.001. Statistical significance was assessed by unpaired t-test with two-tailed calculations. Data is presented as mean ± SEM. (**Q**) qPCR analysis of *Fgf18* expression following *Creb5* siRNA treatment combined with 5 ng/ml TGF-β1 or BSA on E14.5 soft palatal mesenchymal cell culture, compared with the control siRNA. '+' or '-' under the plots indicates the presence or absence of the indicated treatment. *, p≤0.05; **, p≤0.01; ***, p≤0.001. Statistical significance was assessed by ANOVA. Data is presented as mean ± SEM. N=3 for all experiments. *Tgbfr1^{fl/fl}* or *Tgbfr1^{fl/+}* littermates were used as controls for *Osr2^{Cre};Tgfbr1^{fl/fl}* mice. Scale bar in G indicates 100 μm in G-N.

The online version of this article includes the following source data and figure supplement(s) for figure 6:

**Source data 1.** Source data for *Figure 6B*.

**Source data 2.** Source data for *Figure 6C*.

**Source data 3.** Source data for *Figure 6D*.

**Source data 4.** Source data for *Figure 6O*.

**Source data 5.** Source data for *Figure 6P*.

**Source data 6.** Source data for *Figure 6Q*.

**Figure supplement 1.** SCENIC analysis identified individual cell type-specific regulons from E13.5-E15.5 soft palatal scRNAseq data.

**Figure supplement 2.** *Creb5* expression in E13.5-E14.5 *Osr2^{Cre};Tgfbr1^{fl/fl}* and control soft palates.

**Figure supplement 3.** TGF-β signaling, *Creb5*, and *Fgf18* are expressed in a similar region of the perimysial cells of the masseter at E13.5.

have also been used for functional analysis of transcription binding sites (*Stuart et al., 2021*). We designed a CRISPRi plasmid (*Fgf18* CRISPRi Plasmid) whereby the gRNA guides the CRISPRi complex to target only this Smad2/3 binding site, as other binding sites are located distantly, hence blocking its transcriptional activity. Since gRNAs for CRISPRi are reported to have an optimal target effect within a range of 150 bp (*MacLeod et al., 2019*), we examined gRNA targeting sites from +150 bp to -150 bp around the TFBS and identifed a gRNA with the highest targeting efficency and lowest off-target rate, targeting very close to (47 bp downstream of) the TFBS. Next, we transfected the *Fgf18* or control CRISPRi plasmid into the cultured soft palatal mesenchymal cells and compared the response of these cells to TGF-β1 treatment. Since cellular response to TGF-β ligands can be transient rather than accu-mulative (*Sorre et al., 2014*), we first identified the optimal time to analyze *Fgf18* expression levels in the soft palatal mesenchymal cells following TGF-β1 treatment. Consistent with previous studies which found that pSmad2 activity starts to decline around 5 hr following TGF-β1 treatment (*Inman et al., 2002*), we found that 4 hr of TGF-β1 treatment led to a significant increase of Fgf18 expression, which reduced when analyzed 18 hr post-treatment (*Figure 6C*). Thus, 4 hr post-treatment was used as the time point for evaluating the cellular response to the TGF-β1 treatment in the following experiments. Compared with the control CRISPRi plasmid, *Fgf18* CRISPRi plamid significantly attenuated the increase in expresssion of *Fgf18* following TGF-β1 treatment (*Figure 6D*). This finding suggested that the binding to this TFBS in the *Fgf18* promoter region is functionally required for the TGF-β-Smad2/3 signaling cascade to directly activate *Fgf18* expression.

The biological effects of TGF-β signaling are contextual (*Morikawa et al., 2016*). One mechanism for cell-type-specific response to TGF-β signaling is through the cooperation with cell-type-specific master transcription factors, which enable TGF-β signaling to specifically affect the cell-type-specific genes bound by these master regulators (*Mullen et al., 2011*). To identify master regulators that help TGF-β signaling activate a perimysial-specific response in palatal mesenchymal cells, we performed Single-Cell Regulatory Network Interference and Clustering (SCENIC) analysis using E13.5-E15.5

soft palatal scRNAseq data (*Aibar et al., 2017*; *Han et al., 2021*). We identified that gene regulatory networks (GRNs) mediated by *Hic1*, *Creb5*, *Meox1*, *Etv5*, *Tbx15*, *Bach2*, and *Nr2f2* are most associated with perimysial fibroblasts (*Figure 6—figure supplement 1*). The expression patterns of these genes in scRNAseq data indicated that *Hic1*, *Creb5*, *Meox1*, *Etv5*, *Tbx15, and Bach2* are more enriched and specific to the perimysial fibroblasts than other palatal mesenchymal populations, and are thus more likely to be associated with the perimysial fibroblast-specific cell behaviors (*Figure 6E*). Next we explored whether these identfied perimysial factors could interact with TGF-β signaling to activate perimysial-specific TGF-β downstream genes such as *Fgf18*. Among the perimysial-associated factors, Creb5 has been reported to be required for TGF-β signaling to induce *Prg4* expression during chondrocyte development (*Zhang et al., 2021*). *In vivo* expression patterns further validated that, among these regulatory genes (*Figure 6I–N*), *Creb5* was most abundantly expressed in the perimysial fibroblasts (*Figure 6I*), similar to that of *Fgf18* and TGF-β signaling activity (*Figure 6G and H*). Moreover, unlike that of *Tbx15*, *Creb5* expression was present in the palatal mesenchyme as early as E13.5, similar to TGF-β signaling activity and *Fgf18* expression (*Figure 6—figure supplement 2A*). Since its expression did not change in the mutant from E13.5 to E14.5, it is likely to function as a partner rather than a downstream target of the TGF-β signaling (*Figure 6—figure supplement 2A–D*). We thus hypothesized that Creb5 is likely one of the candidates that may function cooperatively with TGF-β signaling to activate perimysial-specific signaling genes such as *Fgf18*. To test the function of Creb5, we first evlauated its role in regulating *Fgf18* expression using E14.5 soft palatal mesenchymal cell culture and found that reduction of *Creb5* in soft palatal mesenchymal cells led to reduction of *Fgf18* expression (*Figure 6O and P*), indicating that Creb5 can regulate *Fgf18* expression. To evaluate the cooperation of Creb5 and TGF-β signaling, we compared the increase of *Fgf18* following TGF-β1 treatment with or without *Creb5* reduction, and found that reducing *Creb5* signficantly attenuated the increase of *Fgf18* expression in soft palatal mesenchymal cells (*Figure 6Q*), suggesting that TGF-β signaling requires cooperation with Creb5 to efficiently activate downstream targets like *Fgf18* in the perimysial fibroblasts. Furthermore, we also found that TGF-β signaling activity (pSmad2), *Creb5* and *Fgf18* are also expressed in a similar pattern in the perimysial cells during the development of the other pharyngeal muscles, such as the first pharyngeal arch-derived masseter muscle (*Figure 6—figure supplement 3*), confirming the possibility that this TGF-β signaling and Creb5 cooperation in regulating perimysial signals, which we identified using the soft palatal muscle model, can plausibly be extended to other craniofacial muscles.

## Fgf18 reduction in perimysial cells leads to reduced myogenic cells in the LVP region of *Osr2^Cre^;Fgf18^fl/fl^* mice while Fgf18 restoration partially rescues myogenic defects in *Osr2^Cre^;Tgfbr1^fl/fl^* samples

To test whether perimysial-derived Fgf18 is functionally required for myogenesis, we generated *Osr2^Cre^;Fgf18^fl/fl^* mice. Unlike in the *Osr2^Cre^;Tgfbr1^fl/fl^* mice, loss of *Fgf18* in the *Osr2^Cre^;Fgf18^fl/fl^* mice did not result in any obvious palatal shelf formation defects compared to the control mice at birth (*Figure 7A–D*), consistent with recent studies demonstrating that Fgf18 is not required for the development of mesenchymal cells themselves in the hard palate (*Xu et al., 2016*; *Yue et al., 2021*). In contrast, the sizes of the soft palatal muscles of the *Osr2^Cre^;Fgf18^fl/fl^* mice appeared to be smaller than those of controls, which was especially obvious in the posterior part of the LVP (*Figure 7G and H*). This was probably due to the difference in muscle fiber organization in the anterior and posterior parts of the LVP. The LVP consists of muscle fibers oriented both parallel and perpendicularly to the section plane in the anterior region (*Figure 7I*) and predominantly perpendicularly in the posterior region (*Figure 7K*). While all the fibers appear more sparse in the anterior and posterior LVP of the *Osr2^Cre^;Fgf18^fl/fl^* mice (*Figure 7J and L*), the perpendicular fibers are significantly reduced (*Figure 7J*). This reduction of LVP muscle size was confirmed by reduced muscle cross-section area (*Figure 7M–Q*) and muscle volume (*Figure 7R–V*) in the *Osr2^Cre^;Fgf18^fl/fl^* mice compared with the controls. To further analyze the progression of muscle defects, we assessed the soft palatal and myogenic phenotypes from E14.5 onwards (*Figure 7—figure supplement 1*). While the MyoD+ myogenic cells appeared comparable in the control and *Osr2^Cre^;Fgf18^fl/fl^* LVP at E14.5 (*Figure 7—figure supplement 1A–E*), the myogenic cells in the *Osr2^Cre^;Fgf18^fl/fl^* mice started to appear progressively reduced compared to the control, and this difference became more apparent at E16.5 (*Figure 7—figure supplement 1F–J*), confirming the role of Fgf18 in late-stage myogenesis. We also confirmed that the loss of functional

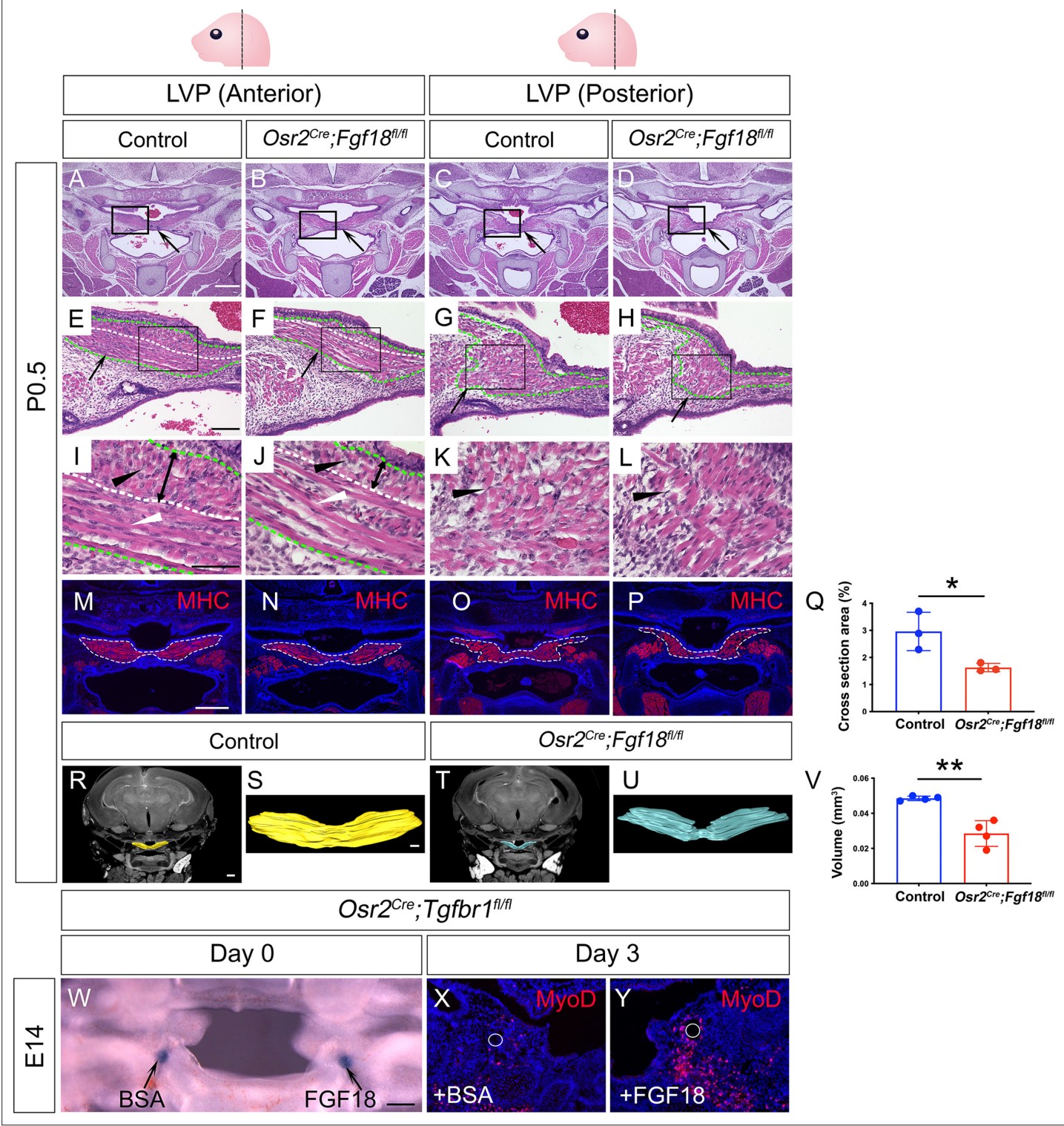

**Figure 7.** Myogenic cells are reduced in the LVP region of *Osr2^Cre^;Fgf18^fl/fl^* mice, while Fgf18 increases MyoD+ myogenic cells in *Osr2^Cre^;Tgfbr1^fl/fl^* soft palate slice cultures. (**A–L**) H&E staining in coronal sections of LVP region at P0.5 from control and *Osr2^Cre^;Fgf18^fl/fl^* mice. Boxed areas in A-D and E-H are enlarged in E-H, and I-L, respectively. Black arrows point to the palatal shelf in A-D and the LVP in E-H. Green dotted line outlined the LVP in E-H. Black and white triangles point to perpendicular muscle fibers in I-L and parallel fibers in I-J. The white dotted line indicates the boundaries of perpendicular and parallel fibers in E, F, I, and J. Double-ended arrows indicate the thickness of perpendicular fibers in I and J. N=4. (**M–Q**) Immunofluorescence and quantification of MHC (red) staining on coronal section of the LVP region of control *Osr2^Cre^;Fgf18^fl/fl^* mice at P0.5. MHC+ areas in the LVP region for quantification of cross-section areas are outlined by white dashed lines in M-P. The percentage of MHC-stained area out of the whole image area is used

*Figure 7 continued on next page*

*Figure 7 continued*

for quantification in Q. *, p≤0.05. N=3. Statistical significance was assessed by unpaired t-test with two-tailed calculations. Data is presented as mean ± SEM. (**R–V**) CT scanning and quantitative analysis of the muscle volume of control and *Osr2^{Cre};Tgfbr1^{fl/fl}* LVP at P0.5. A representative reconstructed control LVP from CT scanning is indicated in yellow (**R and S**) and an *Osr2^{Cre};Tgfbr1^{fl/fl}* reconstructed LVP is indicated in teal (**T and U**). **, p≤0.01. N=4. Statistical significance was assessed by unpaired t-test with two-tailed calculations. Data is presented as mean ± SEM. (**W**) A 300 µm coronal slice of the LVP region at E14 from *Osr2^{Cre};Fgf18^{fl/fl}* mouse for slice culture following bead implantation. Arrows point to BSA- or FGF18-treated bead. (**X–Y**) Immunofluorescence of MyoD (red) in the coronal section of LVP region from *Osr2^{Cre};Fgf18^{fl/fl}* mouse cultured for 3 days with BSA bead (**X**) and FGF18 bead (**Y**). White circles indicate the location of the BSA beads in X and FGF18 beads in Y. N=3. *Fgf18^{fl/fl}* or *Fgf18^{fl/+}* littermates were used as controls for *Osr2^{Cre};Fgf18^{fl/fl}* mice. Scale bars in A, E, I, M, R, S, and W indicate 500 µm in in A-D, 100 µm in E-H, 50 µm in I-L, 400 µm in M-P, 400 µm in R and T, 100 µm in S and U, and 100 µm in W-Y, respectively.

The online version of this article includes the following source data and figure supplement(s) for figure 7:

**Source data 1.** Source data for *Figure 7Q*.

**Source data 2.** Source data for *Figure 7V*.

**Figure supplement 1.** *Osr2^{Cre};Fgf18^{fl/fl}* mice exhibit myogenic defects during LVP development.

**Figure supplement 1—source data 1.** Source data for *Figure 7—figure supplement 1E*.

**Figure supplement 1—source data 2.** Source data for *Figure 7—figure supplement 1J*.

**Figure supplement 2.** Loss of *Fgf18* in *Osr2^{Cre};Fgf18^{fl/fl}* mice leads to defective proliferation of *Myf5+* myogenic cells during LVP development, while exogenous FGF18 can increase the proliferation of C2C12 myogenic cells.

**Figure supplement 2—source data 1.** Source data for *Figure 7—figure supplement 2G*.

**Figure supplement 2—source data 2.** Source data for *Figure 7—figure supplement 2J*.

**Figure supplement 2—source data 3.** Source data for *Figure 7—figure supplement 2K*.

Exon1C of *Fgf18* is efficient in the *Osr2^{Cre};Fgf18^{fl/fl}* soft palatal shelves (*Figure 7—figure supplement 1K–N*). Taken together, these findings suggest that Fgf18 may not be required for determining the fate of palatal mesenchymal cells themselves, but rather may serve as a paracrine molecule for the myogenic cell development.

To further investigate how loss of *Fgf18* led to reduce myogenic cells form E14.5 onwards, we analyzed change of proliferation and cell survival of myogenic cells in *Osr2^{Cre};Fgf18^{fl/fl}* mice compared with the controls. At this stage, the myogenic cells in *Osr2^{Cre};Fgf18^{fl/fl}* mice and controls are rarely apoptotic (*Figure 7—figure supplement 2A and D*), but display proliferative activity in both mutants and controls (*Figure 7—figure supplement 2B and E*). Therefore, we focused on analyzing the proliferative status of the myogenic cells. Since *Myf5* expression was mostly associated with proliferating cells with enriched *Fgfr4* expression, we next compared the proliferative status of the *Myf5+* myogenic progenitors specifically in *Osr2^{Cre};Fgf18^{fl/fl}* and control mice. We could see a significant reduction of proliferation of *Myf5+* cells in *Osr2^{Cre};Fgf18^{fl/fl}* mice (*Figure 7—figure supplement 2C, F, and G*), suggesting that loss of *Fgf18* led to reduced myogenic progenitor proliferation in the *Osr2^{Cre};Fgf18^{fl/fl}* mice, which ultimately led to smaller muscle mass. Furthermore, to confirm that Fgf18 could directly work on myogenic cells, we treated C2C12 mouse myogenic cells, which we maintained as undifferentiated myoblasts in growth medium, with FGF18 and found that the proliferation rate of the C2C12 cells significantly increased in growth medium as early as 1 day later (*Figure 7—figure supplement 2H–J*), eventually leading to significantly increased C12C2 cell numbers at 3 days post-treatment (*Figure 7—figure supplement 2K*), supporting the role of Fgf18 in directly regulating myogenic progenitor proliferation *in vivo*.

To test whether Fgf18 treatment could restore some of the severe muscle formation defects following the loss of TGF-β signaling in *Osr2^{Cre};Tgfbr1^{fl/fl}* mice, we optimized an organ culture system (*Alfaqeeh and Tucker, 2013*; *Humpel, 2015*) to culture 300 µm-thick slices of embryonic head tissues containing the LVP region. This enabled the maintenance of the three-dimensional structure of the pharyngeal region and allowed for efficient nutrient penetration and bead implantation (*Figure 7W*). Using this culture system, we found that FGF18 bead treatment in the soft palatal shelves led to increased MyoD+ cells compared with BSA beads in *Osr2^{Cre};Tgfbr1^{fl/fl}* slice cultures (*Figure 7X and Y*). This suggests that Fgf18 is one of the key regulators of perimysial-to-myogenic signaling affected in the LVP in the *Osr2^{Cre};Tgfbr1^{fl/fl}* mice. Taken together, our studies identified that TGF-β signaling interacts with perimysial regulator Creb5 to specify the perimysial-to-myogenic signaling, such as Fgf18, in individual pharyngeal muscle development.

## Discussion

In the later stage of pharyngeal muscle development, myogenic progenitors migrate into discrete myogenic sites to form individual muscle anlagen anatomically resembling their adult counterparts (*Noden and Francis-West, 2006*; *Sambasivan et al., 2011*; *Shih et al., 2008*; *Ziermann et al., 2018*). How these fine-tuned individual muscles form the right morphology at the right location in order to perform their physiological functions remains largely unknown. Here, using the pharyngeal muscle LVP as a model, our study reveals that an important aspect of this mechanism is the establishment of a distinct pro-myogenic perimysial subdomain adjacent to the myogenic site. The cells in this domain have unique identities and regulatory mechanisms distinct from the rest of connective tissue cell populations within the palatal mesenchyme, enabling perimysial cells to uniquely provide various pro-myogenic signals and define specific myogenic sites to support region-specific myogenesis. Using unbiased screening with scRNAseq analysis combined with mouse genetic approaches, we identified TGF-β signaling as a predominant and specific regulator for perimysial cell fate determination during this stage, and perimysial transcriptional factor Creb5 assists TGF-β signaling to achieve functional specificity in supporting pharyngeal myogenesis via pro-myogenic signals such as Fgf18.

In early myogenesis, CNC cells have also been shown to induce myogenic differentiation by secreting both BMP and Wnt inhibitors to antagonize the dorsal neural tube-derived BMP and Wnt signaling molecules that repress craniofacial skeletal muscle formation (*Tzahor et al., 2003*). In the later developmental stages, when the CNC-derived palatal mesenchymal cells are more differentiated and patterned into distinguishable functional domains, we have the optimal opportunity to explore a specific regulatory mechanism of myogenesis. We found that during the LVP development, the late perimysial population (perimysial fibroblasts) with the most active TGF-β signaling is also distributed according to a muscle-specific pattern in both the fourth pharyngeal arch-derived LVP and first pharyngeal arch-derived masseter muscle; and this TGF-β signaling function is required for the formation of the muscle analgen of both the LVP and masseter (*Han et al., 2014*). Similarly, in limb muscle development, a group of pre-patterned *Tcf4+* limb mesodermal cells also pre-determines the basic pattern of the muscles (*Kardon et al., 2003*). This indicates a potentially universal mechanism in which a pre-defined perimysial domain distinct from the rest of the connective tissues is required to establish specific myogenic sites that allow for proper muscle analge specification in late developmental stages.

By investigating perimysial-to-myogenic communication in this study, we have identified pro-myogenic signaling from the neighboring late perimysial cells (perimysial fibroblasts). It thus appears that these embryonic myogenic progenitors may also require 'embryonic niches' to support their contribution to muscle development at specific myogenic sites. In adults, muscle stem cells – satellite cells – reside in specific niches, and their ability to potentiate muscle repair and regeneration is also supported by signals from these niches (*Andersen et al., 2013*; *Relaix et al., 2021*). By comparing the 'adult muscle stem cell niche' with the 'embryonic niche', more molecular and cellular similarities can be further identified. For example, several perimysial-derived signaling pathways for the embryonic myogenic progenitors are also required for satellite cell fate regulation after birth (*Bigas and Espinosa, 2016*; *Farin et al., 2016*; *Stantzou et al., 2017*). In addition, while *Hic1+* mesenchymal progenitors in the adult niche coordinate various aspects of skeletal muscle regeneration (*Scott et al., 2019*), they are also inferred to be regulators of the perimysial cells of the embryonic niche during development. These similarities suggest potentially conserved mechanisms between embryonic muscle development and adult muscle repair/regeneration, consistent with the clinical observation that patients with disrupted soft palatal muscle development also exhibit impaired differentiation and a reduced number of satellite cells (*Carvajal Monroy et al., 2012*). Thus, the mechanism we identified during development could also be potentially useful for promoting muscle growth after birth.

Craniofacial anomalies such as DiGeorge syndrome, as well as other syndromic and non-syndromic forms of cleft lip and palate, often affect pharyngeal muscle formation and lead to difficulties in eating, facial expression, speaking, and swallowing (*Kelly et al., 2004*; *Kernahan et al., 1984*; *Carvajal Monroy et al., 2012*; *Scambler, 2000*). While the identification of a detailed Pitx2-Tbx1-Msc-Tcf21 transcription factor regulatory network for the pharyngeal myogenic cells may influence the intrinsic regulatory network within the myogenic cells to promote pharyngeal muscle development (*Buckingham and Rigby, 2014*; *Sambasivan et al., 2011*), manipulation through paracrine signaling is a more likely mechanism. In this study, we have identified a variety of potential pro-myogenic signaling molecules from perimysial-to-myogenic cell interaction analysis. A 'chemokine/cytokine cocktail' may

be needed to restore the muscle defects in our cleft soft palate animal model (*Osr2^{Cre}*;*Tgfbr1^{fl/fl}*). As a proof of concept, we found partial improvement of the muscle defects after Fgf18 restoration and thus confirmed the potential of restoring pharyngeal muscle defects with the combination of pro-myogenic factors we identified in the study. Moreover, some of these molecules, including Fgf18 and Dlk1, have also been suggested to be involved in the development and regeneration of various muscles throughout the body including in the limb, diaphragm, and tongue (*Andersen et al., 2013*; *Ito et al., 2018*; *Mok et al., 2014*; *Yue et al., 2021*), extending their potential application to defects in other muscles in the body.

In summary, our study has been able to decipher the molecular and cellular composition of the embryonic myogenic niche for the development of pharyngeal muscles and potentially other muscles in the body. Our work will contribute to a better understanding of the fine-tuned regulatory network of late-stage muscle morphogenesis and lead to the development of novel treatment strategies for regenerating/repairing muscle defects, particularly in infants with birth defects.

# Materials and methods

**Key resources table**

| Reagent type (species) or resource | Designation | Source or reference | Identifiers | Additional information |
|---|---|---|---|---|
| Strain, strain background (*M. musculus*) | *Osr2^{Cre}* | Rulang Jiang, Cincinnati Children's Hospital; *Chen et al., 2009* | | |
| Strain, strain background (*M. musculus*) | *Tgfbr1^{fl/fl}* | *Dudas et al., 2006*; *Larsson et al., 2001* | | |
| Strain, strain background (*M. musculus*) | *Fgf18^{fl/fl}* | David Ornitz, Washington University School of Medicine; *Hagan et al., 2019* | | |
| Strain, strain background (*M. musculus*) | *Rosa26^{LSL-tdTomato}* | Jackson Laboratory; *Madisen et al., 2010* | Stock No. 007905 RRID:IMSR_JAX:007905 | |
| Strain, strain background (*M. musculus*) | C57BL/6J | Jackson Laboratory | Stock No. 000664 RRID:IMSR_JAX:000664 | |
| Sequence-based reagent | RNAScope Probe-Mm-*Aldh1a2* | Advanced Cell Diagnostics | Cat# 447391 | |
| Sequence-based reagent | RNAScope Probe-Mm-*Tbx15* | Advanced Cell Diagnostics | Cat# 558761 | |
| Sequence-based reagent | RNAScope Probe- Mm-*Smoc2* | Advanced Cell Diagnostics | Cat# 318541 | |
| Sequence-based reagent | RNAScope Probe-Mm-*Fgf18* | Advanced Cell Diagnostics | Cat# 495421 | |
| Sequence-based reagent | RNAScope Probe-Mm-*Lama4* | Advanced Cell Diagnostics | Cat# 494901 | |
| Sequence-based reagent | RNAScope Probe- Mm-*Fgfr1* | Advanced Cell Diagnostics | Cat# 443491 | |
| Sequence-based reagent | RNAScope Probe- Mm-*Fgfr2* | Advanced Cell Diagnostics | Cat# 443501 | |
| Sequence-based reagent | RNAScope Probe- Mm-*Fgfr3* | Advanced Cell Diagnostics | Cat# 440771 | |
| Sequence-based reagent | RNAscope Probe- Mm-*Fgfr4* | Advanced Cell Diagnostics | Cat# 443511 | |
| Sequence-based reagent | RNAScope Probe- Mm-*Myf5* | Advanced Cell Diagnostics | Cat# 492911 | |
| Sequence-based reagent | RNAScope Probe-Mm-*Hic1* | Advanced Cell Diagnostics | Cat# 464131 | |
| Sequence-based reagent | RNAScope Probe-Mm-*Creb5* | Advanced Cell Diagnostics | Cat# 572891 | |
| Sequence-based reagent | RNAScope Probe-Mm-*Meox1*-C2 | Advanced Cell Diagnostics | Cat# 530641-C2 | |
| Sequence-based reagent | RNAScope Probe-Mm-*Etv5* | Advanced Cell Diagnostics | Cat# 316961 | |
| Sequence-based reagent | RNAScope Probe-Mm-*Bach2*-C3 | Advanced Cell Diagnostics | Cat# 887121-C3 | |
| Sequence-based reagent | RNAScope Probe-Mm-*Myod1* | Advanced Cell Diagnostics | Cat# 316081 | |

*Continued on next page*

*Continued*

| Reagent type (species) or resource | Designation | Source or reference | Identifiers | Additional information |
|---|---|---|---|---|
| Sequence-based reagent | RNAScope Probe- Mm-*Myod1*-C2 | Advanced Cell Diagnostics | Cat# 316081-C2 | |
| Sequence-based reagent | BaseScope Probe- BA-Mm-*Fgf18*-3zz-st-C1 | Advanced Cell Diagnostics | Cat# 1118021-C1 | |
| Antibody | Mouse monoclonal, Myosin heavy chain (MHC) | DSHB | Cat# MF20 | (1:25) |
| Antibody | Rabbit polyclonal, RFP | Rockland | Cat# 600-401-379 | (1:500) |
| Antibody | Rabbit monoclonal, pSmad2 | Cell Signaling Technology | Cat# 3108 RRID:AB_490941 | (1:500) |
| Antibody | Rabbit monoclonal, MyoD | Abcam | Cat# ab133627 RRID:AB_2890928 | (1:200) |
| Antibody | Rat monoclonal, BrdU | Abcam | Cat# ab6326 RRID: AB_305426 | (1:100) |
| Antibody | Rabbit Polyclonal, Cleaved Caspase-3 | Cell Signaling Technology | Cat# 9661 RRID: AB_2341188 | (1:100) |
| Antibody | Rabbit monoclonal, Smad2/3 | Cell Signaling Technology | Cat# 8685 RRID:AB_10889933 | (1:20) |
| Antibody | Rabbit monoclonal, IgG | Cell Signaling Technology | Cat# 3900 RRID:AB_1550038 | (1:20) |
| Antibody | Goat polyclonal anti-Mouse Alexa Fluor 488 | Life Technologies | Cat# A-11001 RRID:AB_2534069 | (1:200) |
| Antibody | Goat polyclonal anti-Rat Alexa Fluor 488 | Life Technologies | Cat# A-11006 RRID:AB_141373 | (1:200) |
| Antibody | Goat polyclonal anti-Rabbit Alexa Fluor 488 | Life Technologies | Cat# A-11008 RRID:AB_143165 | (1:200) |
| Antibody | Goat polyclonal anti-Rabbit Alexa Fluor 568 | Life Technologies | Cat# A-11036 RRID:AB_10563566 | (1:200) |
| Antibody | Goat polyclonal anti-rabbit IgG Antibody (H+L), HRP | Vector Laboratories | Cat# PI-1000 RRID:AB_2336198 | (1:200) |
| Peptide, recombinant protein | Recombinant human TGF-β1 | R&D Systems | Cat# 7754-BH | |
| Peptide, recombinant protein | Recombinant human FGF18 | Peprotech | Cat# 100–28 | |
| Cell line (*Mus musculus*) | C2C12 | ATCC | Cat# CRL-1772 RRID:CVCL_0188 | |
| Commercial assay or kit | RNAScope Multiplex Fluorescent Kit v2 | Advanced Cell Diagnostics | Cat# 323110 | |
| Commercial assay or kit | BaseScope detection reagent kit v2-Red | Advanced Cell Diagnostics | Cat# 323910 | |
| Commercial assay or kit | TSA Plus Cyanine 3 System | Perkin Elmer | Cat# NEL744001KT | |
| Commercial assay or kit | TSA Plus Fluoresceine System | Perkin Elmer | Cat# NEL771B001KT | |
| Commercial assay or kit | RNeasy Micro Kit | QIAGEN | Cat# 74004 | |
| Commercial assay or kit | iScript Advanced cDNA Synthesis Kit | Bio-Rad | Cat# 1725038 | |
| Commercial assay or kit | SsoFast EvaGreen Supermix | Bio-Rad | Cat# 1725201 | |

*Continued on next page*

*Continued*

| Reagent type (species) or resource | Designation | Source or reference | Identifiers | Additional information |
|---|---|---|---|---|
| Commercial assay or kit | Chromium single-cell 3' v2 reagent kit | 10x Genomics | Cat# PN-120267 | |
| Commercial assay or kit | Cut and Run assay kit | Cell Signaling | Cat# 86652 | |
| Software, algorithm | Cell ranger | 10x Genomics.Inc | RRID:SCR_017344 | |
| Software, algorithm | Seurat | Satija lab | RRID:SCR_016341 | |
| Software, algorithm | CellChat | Jin Lab | | |
| Software, algorithm | SCENIC | Aerts lab | RRID:SCR_017247 | |
| Software, algorithm | GraphPad Prism | GraphPad Software | RRID:SCR_002798 | |

## Animal studies

The *Osr2^Cre* (gift from Rulang Jiang, Cincinnati Children's Hospital; **Chen et al., 2009**), *Rosa26^LSL-tdTomato* (JAX#007905, Jackson Laboratory; **Madisen et al., 2010**), *Tgbfr1^fl/fl* (*Alk5^fl/fl*) (**Dudas et al., 2006**; **Larsson et al., 2001**), *Fgf18^fl/fl* (gift from David Ornitz, Washington University School of Medicine; **Hagan et al., 2019**), and C57BL/6J (JAX#000664, Jackson Laboratory) mouse lines have all been described previously. To generate *Osr2^Cre;Rosa26^LSL-tdTomato* mice, we crossed *Osr2^Cre* male mice with *Rosa26^LSL-tdTomato* female mice. To generate *Osr2^Cre;Tgfbr1^fl/fl* mice, we crossed *Osr2^Cre;Tgfbr1^fl/+* male mice with *Tgfbr1^fl/fl* female mice. To generate *Osr2^Cre;Fgf18^fl/fl* mice, we bred male *Osr2^Cre;Fgf18^fl/+* mice with female *Fgf18^fl/fl* mice. All mice were genotyped using genotyping primers as previously reported. The embryonic samples and newborn pups were collected and used for analysis without consideration of sex. For *in vivo* BrdU labelling, mice were injected intraperitoneally with 10 mg/ml BrdU in PBS, and 100 mg/kg BrdU was administered to each mouse. The mice were then euthanized for proliferation analysis 2 hr after BrdU administration. All animal handling followed federal regulation and was performed with the approval of the Institutional Animal Care and Use Committee (IACUC) at the University of Southern California documented under protocol numbers 9320 and 11765.

## Tissue processing

Embryonic and newborn mouse heads were dissected and fixed in 10% formalin (HT501128, MilliporeSigma) overnight at room temperature following decalcification in 10% EDTA depending on the stage. For paraffin sections, samples were processed in serially ascending concentrations of ethanol solution at room temperature followed by xylene and paraffin wax at 60 °C, then embedded in paraffin wax and sectioned at 8 μm using a microtome (RM2255, Leica). Deparaffinized sections were stained with Hematoxylin and Eosin (H&E). For cryosections, samples were dehydrated in 15% sucrose/PBS solution followed by 30% sucrose/50% Tissue-Tek OCT compound (4583, Sakura). Samples were embedded in the OCT compound and frozen on a block of dry ice. Embedded samples were then cryosectioned at 8 μmusing a cryostat (CM13050S, Leica).

## Immunostaining

Sections were antigen-retrieved for 10 min in preheated antigen unmasking solution (H-3300, Vector Laboratories) followed by 10 min of incubation with 1% Triton X in PBS (T8787, Sigma Aldrich). Sections were then incubated with 1% blocking reagent (PerkinElmer, FP1012) for 1 hr and then primary antibody overnight at 4 °C. The sections were then incubated for 2 hr at room temperature with Alexa Fluor 488 or 568-conjugated secondary antibodies or HPR conjugated antibodies. For TSA-based immunofluorescent staining, the sections were further incubated with 1:200 TSA Plus FITC or Cy3 reagents for 3–5 min of signal development (NEL774001KT or NEL771B001KT, Akoya Bioscience). Sections were then counterstained with 4',6-diamidino-2-phenylindole (DAPI; D9542, Sigma-Aldrich) and mounted with Fluoro-Gel (17985–10, EMS). The sections were washed in phosphate-buffered saline (PBS) with 0.1% Tween 20 (P1379, Sigma) between incubations. The primary and secondary antibodies used in this study were pSmad2 (3108, Cell signaling, 1:500 with TSA), MyoD (ab133627, Abcam; 1:200 with TSA), RFP (600-401-379, Rockland, 1:500 with TSA), myosin heavy chain (MHC)

(MF20, DSHB, 1:25), BrdU (ab6326, Abcam, 1:100), Cleaved Caspase-3 (9661, Cell signaling, 1:200 with TSA), Alexa Fluor 488 anti-mouse (A11001, Thermo fisher Scientific, 1:200), Alexa Fluor 488 anti-rat (A11006, Thermo Fisher Scientific, 1:200), Alexa Fluor 568 anti-mouse (A11004, Thermo Fisher Scientific, 1:200), Alexa Fluor 488 anti-rabbit (A11008, Thermo Fisher Scientific, 1:200), Alexa Fluor 568 anti-rabbit (A11036, Thermo Fisher Scientific, 1:200), and anti-rabbit HRP (PI-1000, Vector Laboratories, 1:200).

## RNAScope *in situ* hybridization (ISH) assay

Tissue sections were air-dried at 60 °C for 1 hr to overnight. RNAScope multiplex fluorescent reagent kit v2 (323100, Advanced Cell Diagnostics) and BaseScope detection reagent kit v2-Red (323910, Advanced Cell Diagnostics) were used for *in situ* hybridization according to the manufacturer's instructions. RNAScope probes from Advanced Cell Diagnostics used in this study were *Aldh1a2* (447391), *Tbx15* (558761), *Smoc2* (318514), *Fgf18* (495421), *Fgfr1* (443491), *Fgfr2* (443501 s), *Fgfr3* (440771), *Fgfr4* (443511), *Lama4* (494901), *Hic1* (464131), *Creb5* (572891), *Meox1* (530641-C2), *Etv5* (316961), *Bach2* (887121-C3), *Myod1* (316081 or 316081-C2), *Myf5* (492911) and *Fgf18-Exon1C* (1118021-C1).

## Single-cell RNAseq

Soft palate tissue (dissected from the posterior third of the palatal region) from E14.5 control and *Osr2^Cre^;Tgfbr1^fl/fl^* embryos was digested and dissociated into single-cell suspension with TrypLE Express enzyme (12605010, Thermo Fisher Scientific) at 37 °C with shaking at 600 rpm for 15 min using a Thermomixer (2231000269, Thermo Fisher Scientific). Single-cell suspension was loaded into the 10x Genomics Chromium system and prepared for single-cell library construction using the 10x Genomics Chromium single-cell 3' v2 reagent kit (PN-120267, 10x Genomics) according to the manufacturer's protocol. Sequencing was performed on Novaseq 6000 (Illumina). Library quality control, sequence alignment, and read counts were analyzed using Cell Ranger 4.0.0. Two control samples were combined using the function *cellranger aggr* so that the combined control sample had comparable cell numbers to the sample from *Osr2^Cre^;Tgfbr1^fl/fl^* mice. One combined control and one *Osr2^Cre^;Tgfbr1^fl/fl^* sample were used for analysis. For each sample, raw read counts from every single cell were analyzed to identify cell clusters and variably expressed genes in each cluster using the Seurat R package (*Hao et al., 2021*) as previously described (*Han et al., 2021*). Seurat was also used to combine the E14.5 control and *Osr2^Cre^;Tgfbr1^fl/fl^* embryos to perform integration analysis as previously described (*Han et al., 2021*). RunPCA and RunUMAP visualization were used for downstream analysis and visualization. The integrated Seurat object combining E13.5-E15.5 control soft palates generated in our previous study (*Han et al., 2021*) was used to analyze gene regulatory network inference and ligand-receptor interactions. Gene regulatory network inference was performed using the R package SCENIC (*Aibar et al., 2017*). Transcription factors for each cell population in the palate were identified using GENIE3 and compiled into regulons, then subjected to cis-regulatory motif analysis. Regulon activity was then scored using AUCell. CellChat (*Jin et al., 2021*) was used to identify the potential ligand-receptor interactions between cell populations in the soft palate. Pre-processing functions (identifyOverExpressedGenes, identifyOverExpressedInteractions, and projectData) and core functions (computeCommunProb, computeCommunProbPathway, and aggregateNet with standard parameters) were applied along with other functions (netVisual_circle, netVisual_bubble, netAnalysis_signalingRole_heatmap netAnalysis_dot, and netAnalysis_river) to determine the senders and receivers.

## Cut and Run assay

Soft palatal tissue of E14.5 C57BL/6J mouse embryos was dissected and digested using Multi Tissue Dissociation Kit 3 (130-110-204, Miltenyi Biotec) at 37 °C with shaking at 600 rpm for 15 min in a Thermomixer (2231000269, Thermo Fisher Scientific), followed by passing through 70 μm Pre-Separation Filters (130-110-204, Miltenyi Biotec) to achieve single-cell suspension solution. The single-cell suspension was processed using a Cut and Run assay kit (86652, Cell Signaling) following the manufacturer's instructions. Five μl of Smad2/3 (8685, Cell Signaling) or IgG (3900, Cell Signaling) were used for each reaction. Standard protocol qPCR reaction was run using SsoFast EvaGreen Supermix (Bio-Rad, 172–5202) on a Bio-Rad CFX96 Real-Time Systems. The sequences of qPCR primers used to detect

enriched DNA fragments near the Smad2/3 binding site at Chr11:33098353–33098362 were GGTG GGGTGACTCAACTGAA (forward) and TTGCGTGGCCTAAGGGTAAG (reverse).

## CRISPRi plasmid generation

gRNAs targeting close to the Smad2/3 binding sites (+150 bp to -150 bp) were identified using IDT guide RNA design tool. A gRNA with the sequence TTTCAGTTGAGTCACCCCAC was selected due to its targeting the proximity (47 bp downstream, 33098286–33098306) to the binding site, high on-target score, and low off-target probability. This gRNA sequence was then cloned to pCas-Guide-Puro-CRISPRi plasmid (GE100083, Origene Technologies) by Origene Technologies to generate a *Fgf18* CRISPRi Plasmid. pCas-Guide-Puro-CRISPRi-Scramble plasmid (GE100084, Origene Technologies) was used as control.

## Soft palatal mesenchymal cell culture and qPCR analysis

Soft palatal tissue of E14.5 C57BL/6J mouse embryos was dissected and digested using Multi Tissue Dissociation Kit 3 (130-110-204, Miltenyi Biotec) at 37 °C with shaking at 600 rpm for 15 min in a Thermomixer (2231000269, Thermo Fisher Scientific), followed by passing through 70 µm Pre-Separation Filters (130-110-204, Miltenyi Biotec) to achieve single-cell suspension solution. The cells were seeded in 24-well plates ($0.5x10^5$ cells/well) with collagen coating solution (125–50, MilliporeSigma) and cultured with DMEM (10566016, Thermo Fisher Scientific)/10% Fetal Bovine Serum (12662029, Thermo Fisher Scientific)/ 1% Penicillin-Streptomycin (15140148, Thermo Fisher Scientific). For CRISPRi plasdmid transfection, each six-well plate well was transfected with 2.5 µg plasmid DNA using Lipofectamine LTX (15338030, Thermo Fisher Scientific) following the manufacturer's instructions. One day after transfection, the cells were cultured in fresh complete medium comprised of 10% Fetal bovine serum (12662029, Thermo Fisher Scientific)/DMEM (10566016, Thermo Fisher Scientific)/1% Penicillin-Streptomycin (15140148, Thermo Fisher Scientific) supplemented with 5 µg/ml puromycin (A1113803, Thermo Fisher Scientific) for 2 days. For siRNA treatment, *Creb5* siRNA (siRNA ID:s107062, 4390816, Thermo Fisher Scientific) and control siRNA (4390844, Thermo Fisher Scientific) were delivered to the cells using Lipofectamine RNAiMAX Transfection Reagent (13778100, Thermo Fisher Scientific) according to the manufacturer's instructions. The cells were then treated with the *Creb5* and control siRNA for 48-72 hr before analysis. For TGFβ1 treatment, recombinant human TGFβ1 (7754-BH, R&D Systems) was used at 5 ng/ml for the TGFβ1 treatment group for 4 hr or 18 hr, while an equal volume of BSA (RB04, R&D Systems) was added to the control group for the same period. RNeasy Plus Micro Kit (Qiagen, 74034) was used to isolate RNA. After RNA isolation, cDNA was transcribed using the iScript Advanced cDNA Synthesis Kit (Bio-Rad, 1725038). qPCR reaction was run using SsoFast EvaGreen Supermix (Bio-Rad, 1725201) or SYBR Select Master Mix (Thermo Fisher Scientific, 4472908) on a Bio-Rad CFX96 Real-Time System. The following primer sequences were obtained from PrimerBank (*Wang et al., 2012*) and used for qPCR reactions: *Creb5* (Forward-AGGATCTTCTGCCGTCTTGAT; Reverse-GCGCAGCCTTCAGTCTCAT), *Fgf18* (Forward-CCTGCACT TGCCTGTGTTTAC; Reverse-TGCTTCCGACTCACATCATCT), and *Gapdh* (Forward-AGGTCGGT GTGAACGGATTTG; Reverse-TGTAGACCATGTAGTTGAGGTCA).

## C2C12 cell culture

C2C12 mouse myoblast cell line (CRL-1772, ATCC) was obtained from ATCC and cultured in growth medium composed of DMEM (11965092, Thermo Fisher Scientific)/10% Fetal bovine serum (12662029, Thermo Fisher Scientific)/1% Penicillin-Streptomycin (15140148, Thermo Fisher Scientific) and maintained at low density to avoid differentiation. After 1–2 passages, C2C12 cells were seeded at 3000 cells/cm$^2$ and cultured for 1 day in growth medium prior to the treatment. For proliferation assay, 500 ng/ml FGF18 (100–28, Peprotech) or an equal volume of BSA (RB02, R&D Systems) were added to C2C12 cells in growth medium for 1–3 days. For BrdU labelling, 10 µM BrdU was added to the medium for 2 hr. Cells were fixed in 4% paraformaldehyde (PFA) for 15 min at RT.

## Embryonic head slice cultures

Embryonic mouse heads were dissected and collected in complete BGJb medium: BGJb Medium (Thermo Fisher Scientific, 12591038) with 10% Fetal Bovine Serum (12662029, Thermo Fisher scientific), 0.1 mg/ml L-ascorbic acid (A4403, Sigma-Aldrich), and 1% Penicillin-Streptomycin (15140148,

Thermo Fisher Scientific). Mouse heads were embedded on ice in a mixture of preheated 20% gelatin (G2500-500G, MilliporeSigma) in BGJB culture medium and sectioned to 300 µm slices using Micro-Slicer Zero 1 (10111, Ted Pella). These slices were placed on cell culture inserts (PICM0RG50, Sigma Aldrich) which were inserted into 6-well culture plates and cultured at 37 °C in a 5% $CO_2$ incubator in complete BGJb medium. For bead implantation, Affi-Gel blue agarose beads (1537302, BioRad) were soaked in 1 µg/µl FGF18 (100–28, Peprotech) or BSA for one hour at 37 °C (*Xu et al., 2016*) and then implanted into the soft palatal shelves of E14 *Osr2^{Cre};Tgfbr1^{fl/fl}* head slices. The samples were collected 3 days after bead implantation.

## Imaging and image analysis

The immunofluorescent and RNAScope *in situ* hybridization fluorescent signals were captured using a Leica DMI3000 B research microscope. The brightfield images were captured using Keyence BZ-X710. Quantitative image analysis was performed by Image J. The same thresholds were set up for measuring the same group of samples. Adjacent sections in the LVP regions were measured and average values across sections were used for analysis. 'Analyze Particles' and 'Measure' functions were used to measure the number of cells and percentage of area of staining out of the whole palate area or whole image area.

## MicroCT analysis

MicroCT scans were performed at the University of Southern California Molecular Imaging Center using a SCANCO µCT50 device. The embryonic heads were scanned with the X-ray source at 70 kVp and 114 µA to generate images at a resolution of 10 µm. Morphometric analysis was performed using the AVIZO 9.1 software package (Visualization Sciences Group). Four biological replicates were performed.

## Statistical analysis

Sample sizes were extrapolated from sample sizes of previously published studies. No randomization or blinding were performed. No samples were excluded from the analysis. For qPCR analysis, 3 technical replicates were used for measuring each sample. GraphPad Prism was used for statistical analysis. All bar graphs display mean ± SEM. Un-paired two-tailed t-test was applied to assess statistical significance between two groups of samples. One-way ANOVA was used for analysis of more than two groups of samples. The chosen level of significance for all statistical tests in this study was $p \leq 0.05$.

## Acknowledgements

We thank Bridget Samuels for the critical reading of the manuscript and Giselle Mejia, Andrea Diaz, and Kimi Nakaki for the schematic illustrations. We acknowledge USC Libraries Bioinformatics Service for assisting with data analysis. The bioinformatics software and computing resources used in the analysis were funded by the USC Office of Research and the Norris Medical Library.

## Additional information

### Funding

| Funder | Grant reference number | Author |
| --- | --- | --- |
| National Institutes of Health | R01 DE012711 | Yang Chai |
| National Institutes of Health | U01 DE028729 | Yang Chai |
| National Institutes of Health | R90 DE022528 | Jifan Feng |

The funders had no role in study design, data collection and interpretation, or the decision to submit the work for publication.

## Author contributions

Jifan Feng, Conceptualization, Data curation, Software, Formal analysis, Validation, Investigation, Visualization, Methodology, Writing - original draft, Writing - review and editing; Xia Han, Data curation, Software, Formal analysis, Validation, Investigation, Visualization; Yuan Yuan, Data curation, Software, Formal analysis, Investigation, Visualization, Methodology; Courtney Kyeong Cho, Siddhika Pareek, Jing Bi, Data curation, Formal analysis, Validation, Visualization; Eva Janečková, Data curation, Formal analysis, Validation, Visualization, Methodology; Tingwei Guo, Data curation, Formal analysis, Validation, Investigation, Visualization, Methodology; Md Shaifur Rahman, Data curation, Validation, Investigation, Visualization; Banghong Zheng, Validation, Investigation, Visualization; Junjun Jing, Data curation, Software, Formal analysis, Validation, Investigation, Visualization, Methodology; Mingyi Zhang, Data curation, Software, Formal analysis, Validation, Visualization, Methodology; Jian Xu, Investigation, Methodology, Writing - review and editing; Thach-Vu Ho, Data curation, Validation, Visualization, Methodology; Yang Chai, Conceptualization, Resources, Formal analysis, Supervision, Funding acquisition, Validation, Investigation, Visualization, Methodology, Writing - original draft, Project administration, Writing - review and editing

## Author ORCIDs

Jifan Feng ⓘ http://orcid.org/0000-0002-9944-2604
Jian Xu ⓘ http://orcid.org/0000-0002-8162-889X
Thach-Vu Ho ⓘ http://orcid.org/0000-0001-6293-4739
Yang Chai ⓘ http://orcid.org/0000-0003-2477-7247

## Ethics

All animal handling followed federal regulation and was performed with the approval of the Institutional Animal Care and Use Committee (IACUC) at the University of Southern California documented under protocol numbers 9320 and 11765.

## Decision letter and Author response

Decision letter https://doi.org/10.7554/eLife.80405.sa1
Author response https://doi.org/10.7554/eLife.80405.sa2

# Additional files

## Supplementary files

• MDAR checklist

## Data availability

The E14.5 control and *Osr2^Cre^;Tgfbr1^fl/fl^* soft palatal single-cell RNAseq data generated in this study have been deposited in Gene Expression Omnibus (GEO) under accession code GSE203035. The single-cell RNAseq data for E13.5-E15.5 soft palate (*Han et al., 2021*) used for analysis in this study have been deposited in GEO under accession code GSE155928.

The following dataset was generated:

| Author(s) | Year | Dataset title | Dataset URL | Database and Identifier |
|---|---|---|---|---|
| Feng J | 2022 | Unbiased screening identifies TGF-β signaling and Creb5 cooperatively regulate FGF18 to control pharyngeal muscle development | https://www.ncbi.nlm.nih.gov/geo/query/acc.cgi?acc=GSE203035 | NCBI Gene Expression Omnibus, GSE203035 |

The following previously published dataset was used:

| Author(s) | Year | Dataset title | Dataset URL | Database and Identifier |
|---|---|---|---|---|
| Han X, Chai Y | 2021 | Transcriptome and chromatin accessiblity analysis of mice soft palate to unreveal the role of Runx2 in regulating palate muscle development | https://www.ncbi.nlm.nih.gov/geo/query/acc.cgi?acc=GSE155928 | NCBI Gene Expression Omnibus, GSE155928 |

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
