## [Editor Report]

The authors bioinformatically analyze previous scRNA-seq datasets of the developing mouse soft palate to identify differential signaling pathway activities in the heterogeneous palatal mesenchyme. Identifying TGF-β signaling pathway activity with the perimysial cells, they hypothesize and test whether TGF-β signaling in the perimysial cells might regulate palatal muscle formation. This paper will be of high interest to developmental biologists interested in the molecular regulation of tissue interactions that occur during mammalian palate morphogenesis.

---

## [Decision Letter]

**Decision letter after peer review:**

Thank you for submitting your article "TGF-β signaling and Creb5 cooperatively regulate Fgf18 to control pharyngeal muscle development" for consideration by *eLife*. Your article has been reviewed by 3 peer reviewers, and the evaluation has been overseen by a Reviewing Editor and Marianne Bronner as the Senior Editor. The reviewers have elected to remain anonymous.

The reviewers have discussed their reviews with one another. The consensus is that the paper has potential but needs better quantitation of the results and additional functional experiments. The full reviews are included below to help with you in revision and the essential revisions are summarized below. We hope you find these comments helpful in revising your manuscript.

Essential revisions:

1. The manuscript needs to better define and quantitatively document the defects in LVP development in the Alk5cko embryos at the cellular level. Figure 3 needs to include a quantitative analysis of possible defects in migration, proliferation, and/or survival of the myogenic progenitor cells during early LVP development from E13 to E14.5. Figure 4 needs iterative clustering to better define the perimysial cell subpopulations. For example, less than 25% of the cells in the current Tbx15+ cluster show Tbx15 expression in the control samples and there was a clear loss of Tbx15-expressing cells in the Alk5cko mutant sample. Iterative clustering analysis is needed to identify/document more specific markers of each subpopulation and how they are affected in the Alk5cko mutant. Subsequently, experiments need to define whether proliferation and/or survival of specific subpopulations were differentially affected.

2. More detailed description and data presentation of the network analysis that identified Creb5 as a potential key regulator need to be provided. The function of Creb5 and how it coordinates with TGFb signaling in palate development need to be experimentally defined.

3. How TGFb signaling regulates Fgf18 expression and the LVP developmental defects in the Fgf18cko embryos need to be better and quantitatively analyzed.

4. Functional studies in myogenic progenitor cells are needed to validate direct signaling interactions from perimysial to myogenic cells.

*Reviewer #1 (Recommendations for the authors):*

A few points:

1. If the authors want to suggest that Smads bind to Fgf18, show that the identified elements work in a transcriptional assay (luciferase or similar). Even that is not very direct, since one can make almost any piece of genomic DNA a "reporter" if adding the correct TF for that site. However, it would at least boost the argument that Smads are controlling Fgf18 expression. Alternatively or in addition, the authors could perform Cut and Run in the Alx5 cko background and show that the element is unbound. Otherwise, it is simply supposition that Smads regulate Fgf18 and thus make this data either dispensable or a discussion point.

2. It is unclear how to fix the problem associated with Creb5, though the authors provide no evidence that Creb5 can regulate Fgf18 expression other than knocking down Creb5. Because this is circumstantial evidence, any statement or discussion about Creb5 being a direct regulator of Fgf18 should be removed. Again, one could probably pick a number of factors that might work with Tgfb signaling. Focusing on Creb5 is fine, though there is no strong evidence to support its role over factors.

3. It is unclear if scRNA-seq was performed one or two times for the mutant sample. I firmly believe that duplicate scRNA-seq for mutants can not only be laborious and costly, but also superfluous. This is scRNA-seq: every cell is different. The authors just need to be very clear on this point (it is clear that the control set was run twice, as they state how the data sets were combined using agro).

*Reviewer #3 (Recommendations for the authors):*

1) Please indicate n-values for each experiment directly in the figure legends.

2) In most of the experiments histological samples are indicated and the results appear clear when a whole population of cells is missing or severely reduced. In other cases (ex. Figure 2, 5, etc.) this is less obvious. To avoid misreading the results or the possibility of assaying peripheral sections in some cases, quantifications of cell counts across several sections are necessary to increase confidence in the results.

3) Figures would be easier to understand for the reader with the addition of schemes depicting the orientation and level of the sections.

4) The sentence in line 307 "Thus, Fgf18 is a direct target for TGF-B signaling in the late perimysial cells" is inferred from bioinformatic analyses, and not functionally validated. Please qualify this affirmation.

5) The reduced size of palatal muscles in Osr2-Cre;Fgf18 fl/fl mice is proposed to be due to muscle fiber disorganization in the LVP. Did the authors analyze changes in proliferation/apoptosis in distinct cell populations of these muscles?

6) In Figure 2 the authors can use a specific marker of perimysial cells in combination with pSMAD2 staining to demonstrate at the cell level activation of TGF-β signaling in this cell population.

7) In the text it is mentioned "TGF-β signaling (pSMAD2) was efficiently reduced in perimysial cells." (Line 198), "showed a significant reduction in Tbx15 expression" (Line 223), however, no precise quantification was performed. Please either re-phrase these sentences or provide quantitative data.

8) In general, there is too much emphasis on the direct interaction between perimysial cells and myogenic progenitors while the role of TGF-β receptors has only been assessed in a descriptive way (Figure 5 – 1). Functional validation in myogenic cells would be required to push this conclusion which remains otherwise correlative with compelling evidence. In this context, given that MyoD is used as a myogenic readout, to what extent is the signaling cascade described patterning vs. commitment to myogenesis vs. differentiation?

9) The specificity of each perimysial subpopulation marker is not clear (Figure 4A-F). Displaying the most differentially expressed genes would be clearer.

10) The differences between the levels of Fgf18 and Lama4 (Figure 5C-D) are not evident. Violin plots are not adapted as the perimysial fibroblasts population is much smaller in the mutant, and there are no quantifications.

11) Figure1: (B) explanation on bottom, "Cell group" "Patterns" to fix ◊ "Patterns" "Signals".

12) Figure3: Either in sentences in Result or Figure Legends, please mention the genotyping of "control".

13) Figure4: (A-F) Please explain the clustering of scRNA-seq. What are pink and green clusters?

14) Figure5: (C,D) Please indicate "control" and "Osr2-Cre;Alk5fl/fl" under each data for easier reading. (G) Myod1 should be in the frame.

15) Figure6: (N) ** and * on the bars should be centered.

16) Figure 7: Typos: Alk5fl/fl and Fgf18fl/fl.

---

## [Author Response]

Essential revisions:1. The manuscript needs to better define and quantitatively document the defects in LVP development in the Alk5cko embryos at the cellular level. Figure 3 needs to include a quantitative analysis of possible defects in migration, proliferation, and/or survival of the myogenic progenitor cells during early LVP development from E13 to E14.5. Figure 4 needs iterative clustering to better define the perimysial cell subpopulations. For example, less than 25% of the cells in the current Tbx15+ cluster show Tbx15 expression in the control samples and there was a clear loss of Tbx15-expressing cells in the Alk5cko mutant sample. Iterative clustering analysis is needed to identify/document more specific markers of each subpopulation and how they are affected in the Alk5cko mutant. Subsequently, experiments need to define whether proliferation and/or survival of specific subpopulations were differentially affected.

We appreciate the suggestion to analyze the more detailed molecular changes in the myogenic and perimysial cells in the *Alk5 cko* mice (*Osr2^Cre^;Tgfbr1^fl/fl^* mice). We have performed the following experiments to address these questions.

1) We analyzed E13.5-E14.5 LVP development in the *Alk5 cko* mice and found cell migration defects occurred at E13.5 and proliferation defects at E14.0 in myogenic cells. Furthermore, no myogenic cells were apoptotic in either control or *Alk5 cko* mice; thus, cell survival was not compromised. We have added the results to Figure 3—Figure supplement 3.

2) To better identify the *Tbx15* negative cell subpoplations that are also affected in the *Alk5 cko* mutant sample, we subsetted and re-clustered the perimysial populations from E13.5-E15.5 soft palatal scRNAseq data, as the perimysial cell subpopulations in this analysis are overall more segregated than in the integrated control and *Alk5 cko* scRNAseq anlysis. By comparing the expression patterns of differentially expressed markers across clusters, we identified *Smoc2* as another perimysial marker that was expressed in the region where *Tbx15* was negative but pSmad2 was positive. More specifically, when comparing with the location of myogenic cells, these two markers showed more clear separation of expression patterns: *Tbx15*+ cells were located predominantly in between the LVP myogenic cells while *Smoc2* expression was present mostly around the periphery of the myogenic site. Consistent with their overlap with pSmad2+ cells, both populations were affected. We added these data to Figure 2—Figure supplement 1 and Figure 4G-H.

3) To identify whether the proliferation and/or survival of *Tbx15+* and *Smoc2+* subpopulations were differentially affected in *Alk5 cko* mice prior to E14.5, we first analyzed their expression patterns at E13.5-E14.0 control and in *Alk5 cko* mice. We found that *Tbx15* was expressed at late stages in the palatal mesenchyme. *Tbx15* was not detectable at E13.5, and only starting to be activated at a very low level by a few cell­s at E14.0. Its expression increased abundantly in the control at E14.5 but failed to increase in the palate of *Alk5 cko*, suggesting that the loss of Tbx15 expression at E14.5 is a result of failure of activating *Tbx15* expression between E14.0-E14.5. In parallel, *Smoc2* was expressed extensively in the palatal mesenchyme at E13.5, and become more patterned to the center of the palatal shelf at E14.0. While *Smoc2* expression was similar between control and *Alk5 cko* mice at E13.5, *Smoc2* expression started to become reduced in the palate in *Alk5 cko* mice at E14.0. Since the proliferation rates of the palatal shelves and the *Smoc2+* cells were not affected at E13.5, and very few cells were apoptotic in the palatal mesenchyme of both the control and *Alk5 cko* mice, we concluded that the loss of *Smoc2* was likely also due to differentiation defects. These results suggest that TGF-β regulates the differentiation of both *Tbx15+* and *Smoc2+* subpopulations, not proliferation or cell survival, despite the timing of the activation of their expression. We added these results to Figure 4—figure supplement 2.

2. More detailed description and data presentation of the network analysis that identified Creb5 as a potential key regulator need to be provided. The function of Creb5 and how it coordinates with TGFb signaling in palate development need to be experimentally defined.

We thank the editor and reviewers for these comments and made the following changes as suggested:

1) We have identified multiple potential regulators associated with perimysial fibroblasts from bioinformatic analyses and chose to test *Creb5* as an example to show how these regulators may work as partners for TGF-β signaling. We picked Creb5 as a most likely candidate because it has recently been reported to support TGF-β signaling to induce *Prg4* expression during chondrocyte development (Zhang et al., 2021). In our study, Creb5’s expression pattern also showed that Creb5 is the most abundantly expressed regulator in the perimysial region at E14.5, overlapping with the pSmad2 and *Fgf18* expression area. Moreover, Creb5 expression was present in the palatal mesenchyme as early as E13.5, similar to TGF-β signaling activity and *Fgf18* expression. This temporal and spatial co-localization of *Creb5*, pSmad2, and *Fgf18* expression domains in combination with the literature led us to hypothesize that Creb5 is the most likely candidate partner for TGF-β signaling-mediated *Fgf18* activation. We have edited the text and added this more detailed description to lines 386-396 on page 18.

2) In this study, we focus on the function of Creb5 in cooperating with TGF-β signaling to induce the expression of *Fgf18* during palate development. We optimized the timing for analyzing TGF-β1 treatment response to 4 hours (instead of 24 hours as used in previous experiments), so that we observed significant increase of *Fgf18* following TGF-β1 treatment. We found that not only Creb5 affected the expression of *Fgf18* by itself, but loss of *Creb5* significantly reduced the increase of *Fgf18* following TGF-β1 treatment, suggesting that Creb5 cooperates with TGF-β signaling to activate *Fgf18* expression in soft palatal mesenchymal cells. Despite the abundant expression of *Creb5* in the palatal shelf mesenchyme from E13.5 to E14.5, the detailed functions of Creb5 in craniofacial development have not been reported in the literature. We plan to generate a *Creb5^fl/fl^* mouse model and thoroughly investigate other roles of *Creb5* in palatal/craniofacial development as an independent project following this work. We have updated the data for Figure 6N-O as the revised Figure 6O-Q.

3. How TGFb signaling regulates Fgf18 expression and the LVP developmental defects in the Fgf18cko embryos need to be better and quantitatively analyzed.

We thank the editor and reviewers for the suggestion to perform more detailed characterization of the regulatory mechanism and *Fgf18 cko* embryo phenotype. We have performed the following experiments to address these questions.

1) To better understand the functional requirement for Smads and define the identified binding site in the *Fgf18* promoter region, we initially generated an *Fgf18* WT promoter-driven luciferase reporter plasmid using 2.5 kb upstream of TSS containing the binding site and an *Fgf18* mutant promoter plasmid in which the binding site was mutated from TCCCAGACAC to TCAAATACAC (Itoh *et al.*, 2019). We transfected the plasmids into NIH3T3 mouse embryonic fibroblast cell line, which is similar to our primary soft palatal mesenchymal cells but with higher transfection efficiency and availability. However, the *Fgf18* WT promoter reporter showed low levels of luciferase activity, so we could not draw a definitive conclusion as to whether the mutant promoter reporter had reduced activity. We thus sought an alternative approach for a transcriptional assay and attempted with the recently developed CRISPRi method (Larson et al., 2013), which has also been used for functional analysis of transcription binding sites (Stuart et al., 2021). We generated a CRISPRi plasmid (*Fgf18* CRISPRi plasmid) to target and block transcriptional activity of only this Smad2/3 binding site of interest, as other binding sites are located more distantly from the targeted region. Following transfection of this plasmid into the soft palatal mesenchymal cells, we found that the TGF-β-induced *Fgf18* expression was impaired in the *Fgf18* CRISPRi-treated cells, compared with the scramble control CRISPRi-treated cells. This data suggested that the Smad2/3 binding to this TF binding site in the *Fgf18* promoter region is functionally required for the TGF-β-Smad2/3 signaling cascade to directly activate *Fgf18* expression. We have added these data to pages 17-18 and Figure 6D.

2) To better quantify the developmental defects of *Fgf18 cko* (*Osr2^Cre^;Fgf18^fl/fl^*) embryos, we first performed quantitative analysis of muscle defects (cross section area and volume) of *Fgf18 cko* and control embryos at newborn stage and confirmed the muscle size was significantly reduced in the *Fgf18 cko* embryos. Next, we analyzed the detailed myogenic defects in the *Fgf18 cko* embryos at earlier stages. We quantified the MyoD+ myogenic cells in the control and *Osr2^Cre^;Fgf18^fl/fl^* LVP at E14.5-E16.5, and found the myogenic cell number in the *Osr2^Cre^;Fgf18^fl/fl^* mice was comparable to that of the control at E14.5, but was significantly reduced at E16.5. More specifically, we found that Fgf18 receptor *Fgfr4* was most enriched in *Myf5+* myogenic cells, possibly associated with proliferative progenitor status. Furthermore, in the *Fgf18 cko* embryos, while the myogenic cell numbers were not affected at E14.5, the proliferation rate of *Myf5*+ myogenic cells was reduced, consistent with the reduced myogenic cell number that occurred at E16.5 and the reduced muscle size at newborn stage. These new results have been added to Figure 7M-V, Figure 5I-P, and Figure 7—figure supplement 1E and J, Figure 7—figure supplement 2A-G.

4. Functional studies in myogenic progenitor cells are needed to validate direct signaling interactions from perimysial to myogenic cells.

We appreciate the suggestion to clarify whether Fgf18 can directly regulate myogenic cell fate, without the influence of perimysial cells. Due to the limited number of myogenic cells obtained from the primary palate, which also have limited growth potential to be expanded, we used C2C12 mouse myogenic cell line to test the direct function of Fgf18 in myogenic cell fate determination. We particularly focused on the proliferation changes because *Fgf18 cko* embryos exhibited proliferation defects in myogenic cells. We found that, consistent with the reduced proliferation of myogenic progenitor cells following the loss of *Fgf18* in the *Fgf18 cko* mice, adding FGF18 to C2C12 cells directly promoted the proliferation of myogenic cells and eventually increased the total number of C2C12 cells. We added these results in Figure 7—Figure supplement 2H-K.

Reviewer #1 (Recommendations for the authors):A few points:1. If the authors want to suggest that Smads bind to Fgf18, show that the identified elements work in a transcriptional assay (luciferase or similar). Even that is not very direct, since one can make almost any piece of genomic DNA a "reporter" if adding the correct TF for that site. However, it would at least boost the argument that Smads are controlling Fgf18 expression. Alternatively or in addition, the authors could perform Cut and Run in the Alx5 cko background and show that the element is unbound. Otherwise, it is simply supposition that Smads regulate Fgf18 and thus make this data either dispensable or a discussion point.

We thank the reviewer for the constructive suggestion. We first generated an *Fgf18* WT promoter-driven luciferase reporter plasmid using 2.5 kb upstream of TSS containing the binding site we identified and an *Fgf18* mutant promoter plasmid in which the binding site was mutated from TCCCAGACAC to TCAAATACAC (Itoh *et al.*, 2019). However, the *Fgf18* WT promoter reporter showed low levels of luciferase activity, so we could not draw a definitive conclusion as to whether the mutant promoter reporter reduced the activity. Since Cut and Run in the *Alx5cko* mice requires collecting a large number of litters, we instead attempted the recently developed CRISPRi method, which have also been used for functional analysis of transcription binding sites by render the binding sites unavailable (Stuart et al., 2021). We generated a CRISPRi plasmid (*Fgf18* CRISPRi plasmid) to target only the Smad2/3 binding site of our interest, as other Smad binding sites are located at a distance from the targeted region. Following transfection of this plasmid into the soft palatal mesenchymal cells, we found that the TGF-β-induced *Fgf18* expression was impaired in the *Fgf18* CRISPRi-treated cells, compared with the scramble control CRISPRi-treated cells. This suggested the functional requirement of Smad2/3 binding to the identified binding sites in activating *Fgf18*. We have added this data in Figure 6D.

2. It is unclear how to fix the problem associated with Creb5, though the authors provide no evidence that Creb5 can regulate Fgf18 expression other than knocking down Creb5. Because this is circumstantial evidence, any statement or discussion about Creb5 being a direct regulator of Fgf18 should be removed. Again, one could probably pick a number of factors that might work with Tgfb signaling. Focusing on Creb5 is fine, though there is no strong evidence to support its role over factors.

We thank the reviewer for pointing this out. We have revised the manuscript to remove any statement or discussion about Creb5 being a direct regulator of *Fgf18*. We agree that Creb5 could be one of multiple factors that could interact with TGF-β signaling, so we toned down language about the “key” role of Creb5 by explaining that we used Creb5 just as a representative example to show how these regulators may work as partners for TGF-β signaling on pages 18-19. We have also reworded our previous statement regarding “the most specific expression patterning the late perimysial cells (Figure 6H)…” to “*Creb5* was most abundantly expressed in the perimysial fibroblasts (Figure 6I)…” on page 18.

3. It is unclear if scRNA-seq was performed one or two times for the mutant sample. I firmly believe that duplicate scRNA-seq for mutants can not only be laborious and costly, but also superfluous. This is scRNA-seq: every cell is different. The authors just need to be very clear on this point (it is clear that the control set was run twice, as they state how the data sets were combined using agro).

The single-cell experiment in this study was performed once for both control and mutant samples. The two controls were combined into one sample and used as such (so that the cell numbers were comparable between control and mutant). We agree with the reviewer that one-time single-cell experiments are adequate to provide valuable bioinformatic information which can be further validated by other experimental approaches. This information has been added to page 34 in the Materials and methods.

Reviewer #3 (Recommendations for the authors):1) Please indicate n-values for each experiment directly in the figure legends.

We thank the reviewer for this comment. We have added n-values for all the experiments in all figure legends as suggested.

2) In most of the experiments histological samples are indicated and the results appear clear when a whole population of cells is missing or severely reduced. In other cases (ex. Figure 2, 5, etc.) this is less obvious. To avoid misreading the results or the possibility of assaying peripheral sections in some cases, quantifications of cell counts across several sections are necessary to increase confidence in the results.

We thank the reviewer for suggesting more quantitative analysis. We have added quantification across sections in the LVP regions to help highlight less obvious effects in all the figures.

3) Figures would be easier to understand for the reader with the addition of schemes depicting the orientation and level of the sections.

We thank the reviewer for this suggestion and have added schematic drawings to indicate the orientation and level of the sections for all the figures.

4) The sentence in line 307 "Thus, Fgf18 is a direct target for TGF-B signaling in the late perimysial cells" is inferred from bioinformatic analyses, and not functionally validated. Please qualify this affirmation.

We thank the reviewer for this valuable insight. To support this conclusion, we have adopted a CRISPRi method (Larson et al., 2013), which has been used recently for functional analysis of transcription binding sites by rendering them unavailable (Stuart et al., 2021). Using a CRISRi plasmid that targeted the identified Smad2/3 binding site in the soft palatal mesenchymal cells, we found that *Fgf18* activation following TGF-β1 treatment was attenuated, supporting the notion that binding to this binding site in the *Fgf18* promoter region is functionally required for TGF-β/Smad2/3 signaling to activate *Fgf18*. We have added this data in Figure 6D and updated the statement accordingly in lines 419-428 on pages 17-18.

5) The reduced size of palatal muscles in Osr2-Cre;Fgf18 fl/fl mice is proposed to be due to muscle fiber disorganization in the LVP. Did the authors analyze changes in proliferation/apoptosis in distinct cell populations of these muscles?

We thank the reviewer for the suggestion to analyze the defects more specifically in myogenic subpopulations. After overall analysis of the phenotype of the *Osr2-Cre;Fgf18^fl/fl^* mice (*Osr2^Cre^;Fgf18^fl/fl^* mice), we concluded that the reduced proliferation rate in myogenic progenitors, leading to smaller pool of these cells, is likely the main defect leading to reduced muscle size in the *Osr2^Cre^;Fgf18^fl/fl^* mice. We analyzed the proliferation/apoptosis changes in the myogenic cells in *Osr2^Cre^;Fgf18^fl/fl^* mice and found reduced proliferation in the *Myf5+* subpopulation, which is likely to be the progenitor population. We added this data as Figure 7—figure supplement 1E and J and figure supplement 2.

6) In Figure 2 the authors can use a specific marker of perimysial cells in combination with pSMAD2 staining to demonstrate at the cell level activation of TGF-β signaling in this cell population.

We thank the reviewer for this suggestion and have co-localized all the perimysial markers we identified in this study (*Aldh1a2*, *Tbx15* and *Smoc2*) with pSmad2 to identify the activation of TGF-β signaling in these cell subpopulations. These results have been incorporated into the revised Figure 2—Figure supplement 1G-L.

7) In the text it is mentioned "TGF-β signaling (pSMAD2) was efficiently reduced in perimysial cells." (Line 198), "showed a significant reduction in Tbx15 expression" (Line 223), however, no precise quantification was performed. Please either re-phrase these sentences or provide quantitative data.

We thank the reviewer for this comment and have re-phrased these sentences as suggested.

8) In general, there is too much emphasis on the direct interaction between perimysial cells and myogenic progenitors while the role of TGF-β receptors has only been assessed in a descriptive way (Figure 5 – 1). Functional validation in myogenic cells would be required to push this conclusion which remains otherwise correlative with compelling evidence. In this context, given that MyoD is used as a myogenic readout, to what extent is the signaling cascade described patterning vs. commitment to myogenesis vs. differentiation?

We thank the reviewer for this suggestion. In addition to the data in Figure 5—Figure supplement 1, we co-localized the Fgf18 receptor, *Fgfr4*, with more specific myogenic markers for early progenitor status and later differentiated status. We found that *Fgfr4* was more associated with *Myf5*+ early myogenic cells which are also proliferative, not the more committed *Myog+*/*Myl1+* myogenic cells. Given the proliferation defects observed in the *Myf5*+ cells in the *Osr2^Cre^;Fgf18^fl/fl^* mice, we focused on the direct role of Fgf18 on C2C12 mouse myogenic cell proliferation, and found that Fgf18 regulates the proliferation of these myogenic cells. We have added these data in Figure 5—Figure supplement 1and Figure 7—figure supplement 2.

9) The specificity of each perimysial subpopulation marker is not clear (Figure 4A-F). Displaying the most differentially expressed genes would be clearer.

We thank the reviewer for this suggestion and performed more detailed marker analysis. Using E13.5-E15.5 control scRNAseq clusters, we have identified *Aldh1a2*, *Smoc2*, and *Tbx15* as perimysial markers with differential expression patterns in vivo. However, for the perimysial subpopulations in the E14.5 control and *Osr2^Cre^;Tgfbr1^fl/fl^* (*Osr2-Cre;Alk5^fl/fl^*) integrated perimysial cell clusters (Figure 4A-C), the expression domains of the above clusters are more overlapping, particularly those of the *Tbx15* and *Smoc2* populations. We speculate that the less segregated gene expression patterns could possibly be attributed to integrating the *Osr2^Cre^;Tgfbr1^fl/fl^* cells to the control cells or to the similarity within the perimysial populations at a particular developmental stage. Although we could not identify more specific markers than the three noted above to further subdivide the population, we were able to obtain sufficient resolution to identify the different perimysial populations at this stage and evaluate their expression changes in the *Osr2^Cre^;Tgfbr1^fl/fl^* mice. We have included these results in Figure 2—figure supplement 1 and Figure 4.

10) The differences between the levels of Fgf18 and Lama4 (Figure 5C-D) are not evident. Violin plots are not adapted as the perimysial fibroblasts population is much smaller in the mutant, and there are no quantifications.

We thank the reviewer for identifying this. We have removed the violin plots and used feature plots instead for better visualization of the differences between the expression levels of *Fgf18* and *Lama4* in control and *Osr2^Cre^;Tgfbr1^fl/fl^* mice (Figure 5C-D).

11) Figure1: (B) explanation on bottom, "Cell group" "Patterns" to fix ◊ "Patterns" "Signals".

We fixed the labelling in the bottom of Figure 1B as suggested.

12) Figure3: Either in sentences in Result or Figure Legends, please mention the genotyping of "control".

We appreciate this suggestion for clarification of the genotyping. We have added descriptions of the controls in all the figure legends where controls are mentioned.

13) Figure4: (A-F) Please explain the clustering of scRNA-seq. What are pink and green clusters?

We updated the figure and removed the pink and green clusters/labelling in A-F to show the whole population as an entirety.

14) Figure5: (C,D) Please indicate "control" and "Osr2-Cre;Alk5fl/fl" under each data for easier reading. (G) Myod1 should be in the frame.

We thank the reviewer for the careful evaluation of our data. We updated panels C and D and have added “control” and “*Osr2^Cre^;Tgfbr1^fl/fl^*” (revised nomenclature for “*Osr2-Cre;Alk5^fl/fl^*”) labels to C-E. We also moved “*Myod1*” in frame.

15) Figure6: (N) ** and * on the bars should be centered.

We centered the ** and * on the bars as suggested.

16) Figure 7: Typos: Alk5fl/fl and Fgf18fl/fl.

We appreciate the reviewer’s attention to details. We fixed the typos in Figure 7.